

**Estimating global surface ammonia concentrations inferred**
**from satellite retrievals**
Lei Liu [a, b] , Xiuying Zhang [a, *], Anthony Y.H. Wong [b], Wen Xu [c], Xuejun Liu [c], Yi Li [d],
Huan Mi [e, b], Xuehe Lu [a], Limin Zhao [a], Zhen Wang [a], Xiaodi Wu [a, f]
[a] International Institute for Earth System Science, Nanjing University, Nanjing,
210023, China
[b] Department of Earth and Environment, Boston University, Boston, Massachusetts,
USA
[c] College of Resources and Environmental Sciences, Centre for Resources,
Environment and Food Security, Key Lab of Plant-Soil Interactions of MOE, China
Agricultural University, Beijing, 100193, China
[d] Sunset CES Inc., Beaverton, OR, 97008, USA
[e] College of Surveying and Geo-Informatics, Tongji University, 1239 Siping Road,
Shanghai, China
[f] Jiangsu Center for Collaborative Innovation in Geographical Information Resource
Development and Application, Nanjing, 210023, China
* Correspondence to Xiuying Zhang (zhangxy@nju.edu.cn).
**Abstract**
Ammonia ($NH_3$), as an alkaline gas in the atmosphere, can cause direct or indirect
effects on the air quality, soil acidification, climate change as well as human health.
Estimating surface $NH_3$ concentrations is critically important for modelling the dry
deposition of $NH_3$, which has important impacts on the natural environment. However,
sparse monitoring sites make it challenging and difficult to understand the global
distribution of surface $NH_3$ concentrations both in time and space. We estimated the
global surface $NH_3$ concentrations for the years of 2008-2016 using the satellite $NH_3$
retrievals combining its vertical profiles from the GEOS-Chem. The accuracy



assessment indicates that the satellite-based approach has achieved a high predictive
power for annual surface $NH_3$ concentrations compared with the measurements of all
sites in China, US and Europe ($R^2$=0.76 and RMSE=1.50 µg N m$^{-3}$). The
satellite-derived surface $NH_3$ concentrations had higher consistency with the
ground-based measurements in China ($R^2$=0.71 and RMSE=2.6 µg N m$^{-3}$) than the US
($R^2$=0.45 and RMSE=0.76 µg N m$^{-3}$) and Europe ($R^2$=0.45 and RMSE=0.86 µg N m$^{-3}$)
at a yearly scale. Annual surface $NH_3$ concentrations higher than 6 µg N m$^{-3}$ are
mainly concentrated in the North China Plain of China and Northern India, followed
by 2-6 µg N m$^{-3}$ mainly in southern and northeastern China, India, western Europe
and eastern United States (US). High surface $NH_3$ concentrations were found in the
croplands in China, US and Europe, and surface $NH_3$ concentrations in the croplands
in China were approximately double than those in the croplands in the US and Europe.
The linear trend analysis shows that an increase rate of surface $NH_3$ concentrations
(>0.2 µg N m$^{-3}$ y$^{-1}$) appeared in the eastern China during 2008-2016, and a middle
increase rate (0.1-0.2 µg N m$^{-3}$ y$^{-1}$) occurred in northern Xinjiang over China. $NH_3$
increase was also found in agricultural regions in middle and eastern US with an
annual increase rate of lower than 0.10 µg N m$^{-3}$ y$^{-1}$. The satellite-derived surface $NH_3$
concentrations help us to determine the $NH_3$ pollution status in the areas without
monitoring sites and to estimate the dry deposition of $NH_3$ in the future.

## Introduction

Ammonia ($NH_3$), emitted primarily by agricultural activities and biomass burning, is
an important alkaline gas in the atmosphere (Van Damme et al., 2018;Warner et al.,
2017). Excessive surface $NH_3$ concentrations can cause chronic or acute damage to
the plant (such as reduced growth and bleached gray foliage) when its capacity of
detoxification is exceeded (Eerden, 1982;Sheppard et al., 2008). Estimation of surface
$NH_3$ concentrations is critically important in modelling the dry deposition of $NH_3$,
which may comprise a large part of atmospheric nitrogen (N) deposition, and could
cause acidification in the soil, eutrophication in the aquatic ecosystems, and



contamination in drinking water (Basto et al., 2015;Kim et al., 2014;Lamarque et al.,
2005;Larssen et al., 2011;Reay et al., 2008). In addition, $NH_3$ can also react with
nitric acid and sulfuric acid to form ammonium salts (Li et al., 2014;Li et al., 2017b),
which are important components of particulate matters (PM), and have negative
impacts on air quality and human health (Xu et al., 2017;Schaap et al., 2004).

Several national monitoring programs have been established to quantify surface $NH_3$
concentrations, including the Chinese Nationwide Nitrogen Deposition Monitoring
Network (NNDMN) established in 2004, the Ammonia Monitoring Network in China
(AMoN-China) established in 2015 in China, the Ammonia Monitoring Network in
the US (AMoN-US) as well as the European Monitoring and Evaluation Programme
(EMEP). However, there are still relatively large uncertainties of estimating global
surface $NH_3$ concentrations, resulting from the sparse monitoring sites as well as the
limited spatial representativeness (Liu et al., 2017b;Liu et al., 2017a). Satellite $NH_3$
retrievals are an important complement to gain the global distribution of $NH_3$
concentrations with a high spatial resolution (Van Damme et al., 2014c). $NH_3$ can be
measured by several satellite instruments including the Infrared Atmospheric
Sounding Interferometer (IASI), Atmospheric Infrared Sounder (AIRS), Cross-track
Infrared Sounder (CrIS) and Tropospheric Emission Spectrometer (TES). TES using
the thermal infrared spectral range has sparser spatial coverage compared to IASI,
CrIS and AIRS (Shephard et al., 2011;Zhang et al., 2017a). A recent study (Kharol et
al., 2018) reported the dry $NH_3$ depositions in North America, and found -15%
underestimation in CrIS surface $NH_3$ concentrations (using three fixed $NH_3$ profiles
considering unpolluted, moderate and polluted conditions) compared with the
measurements from the AMoN-US during the warm months (from April to
September). Warner et al. reported the global AIRS $NH_3$ concentrations at 918hPa
(approximately 700-800 m) at $1°$ latitude $\times$ $1°$ longitude grids, and found $NH_3$
concentrations increased in the major agricultural regions during 2003-2015 (Warner
et al., 2017). The IASI $NH_3$ measurements have been validated with $NH_3$ columns
measured by the Fourier transform infrared spectroscopy (FTIR), ground-based $NH_3$



measurements, $NH_3$ emissions and atmospheric chemistry transport models (CTMs)
(Dammers et al., 2016;Van Damme et al., 2014c;Van Damme et al., 2014a;Whitburn
et al., 2016).
Apart from satellite retrievals, CTMs are also powerful tools to investigate
spatiotemporal variability of surface $NH_3$ concentrations in the atmosphere. Schiferl et
al. evaluated the modelled $NH_3$ concentrations during 2008-2012 from GEOS-Chem,
and found an approximately 26% underestimation compared with the ground-based
measurements, which can be related to the relatively large uncertainties in $NH_3$
emissions used for driving GEOS-Chem (Schiferl et al., 2015). Zhu et al. used the
GEOS-Chem constrained by TES measurements to estimate surface $NH_3$
concentration during 2006-2009, and found an improvement in comparison with the
ground-based measurements in the United States (Zhu et al., 2013). Schiferl et al.
used the airborne observations to validate the simulated $NH_3$ concentrations in 2010
from GEOS-Chem, and revealed reasonably simulated $NH_3$ vertical profiles compared
with the aircraft measurements but with an underestimation in surface $NH_3$
concentrations in California (Schiferl et al., 2014). A number of previous studies have
used satellite $NO_2$ columns to estimate the surface $NO_2$ concentrations combining
$NO_2$ vertical profiles from CTMs (Geddes et al., 2016;Lamsal et al., 2013;Nowlan et
al., 2014;Liu et al., 2017c). The methods of using the vertical profiles to convert
satellite-retrieved columns to surface concentrations have been proven successful for
$SO_2$ and $NO_2$ (Geddes et al., 2016;Geng et al., 2015;Lamsal et al., 2008;Nowlan et al.,
2014). CTMs can provide valuable information of $NH_3$ vertical profiles (Whitburn et
al., 2016;Liu et al., 2017b), and IASI-derived surface $NH_3$ concentrations combining
$NH_3$ vertical profiles from CTMs in China and Europe were evaluated previously (Liu
et al., 2017b;Graaf et al., 2018). This study followed these studies to estimate the
satellite-derived global surface $NH_3$ concentrations using IASI $NH_3$ retrievals and the
vertical profiles from GEOS-Chem, and the present study aims to estimate the global
surface $NH_3$ concentration from a satellite perspective.





**Data and Methods**
**IASI NH$_3$ measurements**
The Infrared Atmospheric Sounding Interferometer (IASI) is a passive instrument
measuring infrared radiation within the spectral range of 645-2769 cm$^{-1}$. The IASI-A
instrument is on board of the MetOp-A satellite launched in 2006 covering the globe
twice a day with an elliptical spatial resolution of approximately 12 by 12 kilometers,
and cross the equator at 09:30 and 21:30 local times (Van Damme et al., 2014b). We
used the daytime IASI NH$_3$ measurements due to the larger positive thermal contrast
detected by satellite instruments leading to smaller errors compared to the nighttime
data (Van Damme et al., 2014b). In this work, we used the IASI NH$_3$ columns
products (ANNI-NH3-v2.2R-I) during 2008-2016 (Van Damme et al., 2017) to
estimate the global surface NH$_3$ concentrations. The ANNI-NH3-v2.2R-I datasets
were developed by converting spectral HRI (hyperspectral range index) to NH$_3$
columns through an Artificial Neural Network for IASI (ANNI) algorithm (Whitburn
et al., 2016). This algorithm considered the influence of the NH$_3$ vertical profiles,
pressure, humidity and temperature profiles. The NH$_3$ vertical profile information
used to generate the ANNI NH$_3$ columns were retrieved from GEOS-Chem, which
integrates H$_2$SO$_4$-HNO$_3$-NH$_3$ aerosol thermodynamics mechanism (Whitburn et al.,
2016;Van Damme et al., 2017). The IASI NH$_3$ columns used in this study were
processed into the monthly data at 0.25 $^\circ$ latitude $\times$ 0.25 $^\circ$ longitude grids by the
arithmetic averaging method (Van Damme et al., 2017;Whitburn et al., 2016;Liu et al.,
2017a).

**Surface NH$_3$ measurements**
To evaluate our satellite-derived global surface NH$_3$ concentrations, we collected
available surface NH$_3$ measurements on a regional scale in 2014. In China, we used
the national measurements from the Chinese Nationwide Nitrogen Deposition
Monitoring Network (NNDMN). Surface NH$_3$ concentrations in the NNDMN were
measured by both ALPHA (Adapted Low-cost, Passive High Absorption) and DELTA



(Denuder for Long-Term Atmospheric sampling) systems. The detailed descriptions
on the NNDMN have been described in a previous study (Xu et al., 2015). In the US,
we used the measurements from the AMoN-US, downloaded from the website:
http://nadp.sws.uiuc.edu/AMoN/. Surface $NH_3$ concentrations in the AMoN-US were
measured by the radiello diffusive sampler (http://www.radiello.com) as a simple
diffusion-type sampler collected every 2 weeks (Li et al., 2016). We calculated annual
surface $NH_3$ concentrations by averaging all the measurements (every 2 weeks), and
then used them to compare with satellite-derived surface $NH_3$ concentrations. In
Europe, we used the measurements from the EMEP network
(https://www.nilu.no/projects/ccc/emepdata.html). The EMEP is composed of
multiple national networks in Europe, thus the measured systems differs among
different national networks.

## GEOS-Chem model

We used GEOS-Chem version 11-01 as the chemical transport model to calculate
global $NH_3$ vertical profiles (using the year of 2014 as a case study in the results and
discussion). It has a spatial resolution of $2°$ latitude $\times 2.5°$ longitude $\times 47$ vertical
layers spanning over Earth's surface and about 80 km above it. It is driven by the
meteorological field data of the GEOS-FP (forward-processing) products, which were
produced by NASA GMAO (Global Modelling and Assimilation Office)
(https://gmao.gsfc.nasa.gov/). Here we modelled the $NH_3$ vertical profiles using
GEOS-Chem, and used the monthly averages for analysis. The global $NH_3$ emissions
in GEOS-Chem are based on the EDGAR (Emissions Database for Global
Atmospheric Research) v4.2 (http://edgar.jrc.ec.europa.eu/overview.php?v=42), while
the regional emissions are replaced with MIX inventory for East Asia (Li et al., 2017a)
(http://www.meicmodel.org/dataset-mix.html), EMEP inventory for Europe
(http://www.emep.int/), NEI (National Emissions Inventory, 2011) for the US
(https://www.epa.gov/air-emissions-inventories) and CAC (Criteria Air Contaminant)
inventory for Canada (http://www.acrd.bc.ca/criteria-air-contaminants). The biomass



burning emissions are from Fire INventory from NCAR version 1.0 (FINNv1)
including agricultural fires, wildfire and pre-scribed burning (Wiedinmyer et al.,
2011). The GEOS-Chem simulates a comprehensive atmospheric
$NO_x$-$O_3$-VOC-aerosol system (Mao et al., 2013). The thermodynamic equilibrium of
$NH_3$-$H_2SO_4$-$HNO_3$ system is simulated by the ISORROPIA II model (Fountoukis and
Nenes, 2007;Pye et al., 2009). The modelling of wet deposition is described by a
previous study (Liu et al., 2001) with updates from the studies (Amos et al.,
2012;Wang et al., 2011). Dry deposition of particles follows the size-segregated
treatment (Zhang et al., 2001) and gaseous dry deposition follows the framework
(Wesely, 1989) with updates from a previous study (Wang et al., 1998). We archive
the output daily averages of $NH_3$ concentrations as well as the averages between 9 and
10 am, which corresponds to the local crossing time of IASI (9:30 am). The
relationship between $NH_3$ concentration at 9-10 am and the daily averages derived
from the GEOS-Chem was used to convert the satellite observed $NH_3$ column to daily
averages (Nowlan et al., 2014).

**Estimation of surface $NH_3$ concentrations**

We estimated global surface $NH_3$ concentrations using the IASI $NH_3$ columns as well
as the GEOS-Chem. We took into account the advantages of IASI $NH_3$ columns with
high spatial resolutions and the GEOS-Chem with vertical profiles. The GEOS-Chem
outputs include 47 layers, which are not continuous in the vertical direction. To gain
the continuous vertical $NH_3$ profile, we used the Gaussian function to fit the 47 layers'
$NH_3$ concentrations. The height of each grid box used here was calculated at the
middle height of each layer rather than the top height of each layer. A three-parameter
Gaussian function was used to fit $NH_3$ vertical profiles at each grid box from the
GEOS-Chem in the previous studies (Whitburn et al., 2016;Liu et al., 2017b) :
$\rho = \rho_{max} e^{-(\frac{Z-Z_0}{\sigma})^2}$     (1)
where $\rho$ is $NH_3$ concentrations at the layer height $Z$; $\rho_{max}$ is the maximum $NH_3$
concentrations at the height $z_0$; $\sigma$ is an indicator for the spread or thickness of the

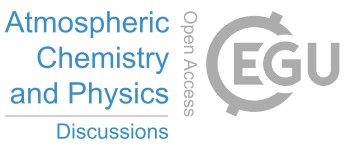

$NH_3$ concentrations.
This study expanded the equation (1) to fit $NH_3$ vertical profiles at each grid box by
the following equation (Liu et al., 2017b):
$\rho = \sum_{i=1}^{n} \rho_{max,i} e^{-(\frac{z-z_{0,i}}{\sigma_i})^2}$    (2)
where n ranges from 1 to 6. If n equals 1, the equation (2) is the same as the equation
(1); if n is greater than 1, the equation (2) is the multiple three-parameters Gaussian
items. We determined the value of n that can simulate the $NH_3$ vertical profiles with
best performance at each grid box using the determining coefficients of R-Square ($R^2$).
Once the $NH_3$ vertical profiles were determined at each grid box, we can extrapolate
$NH_3$ concentrations at any height from the GEOS-Chem ($G_{GEOS-Chem}$).
We then aggregated the IASI $NH_3$ columns $\Omega_{IASI}$ (0.25 ° latitude $\times$ 0.25 ° longitude)
to the GEOS-Chem grid size $\overline{\Omega_{IASI}}$ (2 ° latitude $\times$ 2.5 ° longitude) by the averaging
method. We have the following equation (Lamsal et al., 2008):
$\overline{G_{IASI_{9-10}}} = \frac{G_{GEOS-Chem}}{\Omega_{GEOS-Chem}} \times \overline{\Omega_{IASI_{9-10}}}$    (3)
where $\overline{G_{IASI_{9-10}}}$ is the satellite-derived surface $NH_3$ concentrations at a GEOS-Chem
grid size at 9-10am; $\frac{G_{GEOS-Chem}}{\Omega_{GEOS-Chem}}$ is the ratio of surface $NH_3$ concentrations to $NH_3$
columns calculated from GEOS-Chem; $\overline{\Omega_{IASI_{9-10}}}$ is the average IASI $NH_3$ columns
at a GEOS-Chem grid at 9-10am.
We found a high correlation (R=0.96 and p=0.000) between the surface $NH_3$
concentrations and $NH_3$ columns based on the GEOS-Chem outputs (**Fig. S1**). Then
we used the satellite-derived scaling factor to downscale the satellite-derived surface
$NH_3$ concentrations at a GEOS-Chem grid by using the following ratio:
$R_{IASI} = \frac{\Omega_{IASI}}{\overline{\Omega_{IASI}}}$    (4)
$G_{IASI_{9-10}} = \overline{G_{IASI_{9-10}}} \times R_{IASI}$    (5)
where $R_{IASI}$ is the scaling factor. $G_{IASI_{9-10}}$ is the satellite-derived surface $NH_3$
concentrations at a satellite IASI grid size (0.25 ° latitude $\times$ 0.25 ° longitude) at 9-10am.
To convert the instantaneous satellite-derived surface $NH_3$ concentrations $G_{IASI}$ to



daily average surface $NH_3$ concentrations, we followed the methods (Nowlan et al.,

233    2014):

$$G_{IASI}{}^* = \frac{G_{GEOS-Chem}^{1-24}}{G_{GEOS-Chem}^{9-10}} \times G_{IASI_{9-10}} \quad (6)$$
where $G_{IASI}{}^*$ is the daily average surface $NH_3$ concentrations, and $\frac{G_{GEOS-Chem}^{1-24}}{G_{GEOS-Chem}^{9-10}}$ is the
ratio of the GEOS-Chem surface $NH_3$ concentrations at the daily average to the
average of 9-10 am.

**Results and Discussion**
**$NH_3$ vertical profiles from GEOS-Chem**
$NH_3$ emitted from the surface can be transported horizontally or vertically, and its
concentrations may show a certain gradient in the vertical and horizontal directions (E.
et al., 1997;Rozanov et al., 2005). There are generally two types of shapes of $NH_3$
vertical profiles (**Fig. S2**) from aircraft measurements (Li et al., 2017b;Tevlin et al.,
2017) and CTMs (Whitburn et al., 2016;Liu et al., 2017b). One is representative for
the vertical profile with maximum $NH_3$ concentrations at a certain height ($z_0 > 0$) and
the other is representative for the vertical profile with maximum $NH_3$ concentrations
near the earth surface ($z_0 = 0$). In this study, the vertical profiles of $NH_3$ were fitted
based on the 47 layers' outputs by GEOS-Chem in 2014 at a monthly scale. **Fig. S3**
shows the spatial distribution of $NH_3$ concentrations in the first and fifth layers
simulated by GEOS-Chem in January 2014. $NH_3$ concentrations in the fifth layer are
significantly lower than those in the first layer, suggesting that $NH_3$ concentrations
decrease with increasing layers (or altitude), especially in $NH_3$ hotspot regions (such
as eastern China, India, western Europe and eastern US). The average difference of
$NH_3$ concentrations between the first and fifth layers on the land is 0.34 μg N m$^{-3}$. The
average $NH_3$ concentrations in the first and fifth layers in eastern China, India,
western Europe and eastern US were 2.76, 7.28, 0.55 and 0.31 μg N m$^{-3}$, respectively.

To more vividly depict the vertical profiles of $NH_3$, we show $NH_3$ vertical
concentrations with cross-section drawn at 37$^o$N in January, 2014 (**Fig. S4**). High $NH_3$





concentrations are mainly concentrated in the 1-10 layers, and show a significant
decrease trend with the increasing altitude, which is consistent with the aircraft
measurements (E. et al., 1997;Lin et al., 2014;Levine et al., 1980;Shephard and
Cady-Pereira, 2015;Li et al., 2017b;Tevlin et al., 2017). $NH_3$ vertical profiles were
fitted by Gaussian function (2-6 terms) based on the 47 layers' $NH_3$ concentrations
from the GEOS-Chem, and the fitting accuracy was determined by $R^2$. We found that
the $NH_3$ vertical profiles on the land between 60 °N and 55 °S can be well modelled
using Gaussian function ($R^2$ higher than 0.90) (**Fig. 1**). Previous studies also found
high accuracy using the Gaussian function to simulate the $NH_3$ vertical profiles in
China and globally (Whitburn et al., 2016;Liu et al., 2017b).

## Validation of satellite-derived surface $NH_3$ concentrations

$NH_3$ vertical profiles were used to convert IASI $NH_3$ columns to surface $NH_3$
concentrations. **Fig. 2** shows the IASI-derived global surface $NH_3$ concentrations on
the land at 0.25 ° latitude $\times$ 0.25 ° longitude grids in 2014. IASI-derived surface $NH_3$
concentrations capture the general spatial pattern of surface $NH_3$ concentrations fairly
well in 2014 in regions with relatively intensive monitoring sites ($R^2$=0.76 and
RMSE=1.50 μg N m$^{-3}$ in **Fig. 2 and Fig. 3**). The overall mean of satellite-derived
surface $NH_3$ concentrations in 2014 at the measured sites was 2.52 μg N m$^{-3}$ and was
close to the average of measured surface $NH_3$ concentrations (2.51 μg N m$^{-3}$) in 2014.
IASI-derived surface $NH_3$ concentrations gained higher consistency with the
ground-based measurements in China ($R^2$=0.71 and RMSE=2.6 μg N m$^{-3}$) than the US
($R^2$=0.45 and RMSE=0.76 μg N m$^{-3}$) and Europe ($R^2$=0.45 and RMSE=0.86 μg N m$^{-3}$)
at a yearly scale. This might be due to the fact that for high concentrations in a region
(associated with high thermal contrast) can be more reliably detected by IASI (Van
Damme et al., 2014a). Similarly, we also compared the surface $NH_3$ concentrations (at
the first layer) simulated by GEOS-Chem with the monitoring results ($R^2$=0.54 and
RMSE=2.14 μg N m$^{-3}$ in **Fig. 3**). In general, IASI-derived surface $NH_3$ concentrations
had better consistency with the ground-based measurements than those from





GEOS-Chem over China, the US and Europe. The relatively low accuracy from
GEOS-Chem was likely due to the coarse model resolutions as well as the poor
spatiotemporal representations of $NH_3$ emissions, as suggested by a previous study
(Zhang et al., 2018).

A known limitation of IASI $NH_3$ retrievals is lack of the vertical profile information.
A previous study (Van Damme et al., 2014a) used the fixed profiles on the land to
convert the IASI $NH_3$ columns to surface $NH_3$ concentrations. Using the fixed profiles
can cause large uncertainties for estimating surface $NH_3$ concentrations. In this work,
we utilized the advantages of CTMs and considered the spatial variability of the
vertical profiles, and proves that IASI $NH_3$ columns are powerful to predict the
surface $NH_3$ concentrations combining the vertical profiles simulated by Gaussian
function.

Through the Gaussian simulation of $NH_3$ vertical profiles, we are able to evaluate the
sensitive regions of surface $NH_3$ concentrations with respect to different heights. **Fig.**
**S5** shows the spatial distribution of the difference of $NH_3$ concentrations between
40m and 60m (about the middle height of the first layer in GEOS-Chem). In general,
in strong $NH_3$ emission regions, there is a relatively large difference in surface $NH_3$
concentrations such as, for instance, in eastern China and northwestern India (can be
up to 3 µg N m$^{-3}$); subsequently, a middle difference (2-3 µg N m$^{-3}$) occurs in eastern
and middle China, northern India and northern Italy. Except above mentioned regions,
the difference of $NH_3$ concentrations between 40m and 60m is generally lower 0.5 µg
N m$^{-3}$.

**Spatial distributions of satellite-derived surface $NH_3$ concentrations**
**Fig. 4** shows the spatial distributions of surface $NH_3$ concentrations in China, US and
Europe in 2014. The overall mean surface $NH_3$ concentrations over China were 2.38
µg N m$^{-3}$, with the range of 0.22-13.11 µg N m$^{-3}$. We found large areas in eastern



China, Sichuan Basin and northwestern Xinjiang with surface $NH_3$ concentrations
greater than 8 µg N $m^{-3}$ $y^{-1}$, which were in agreement with the spatial distributions of
the croplands in China (**Fig. S6**). It is not surprising that high surface $NH_3$
concentrations occurred in eastern China and Sichuan Basin because the major
Chinese croplands are distributed there, as the major source of $NH_3$ emissions with
frequent N fertilizer applications. Overall, there was a significant linear correlation
between surface $NH_3$ concentration and N fertilization in China ($R^2$=0.65, p=0.000 in
**Fig. 5**). The hotspots also occurred in northwestern Xinjiang surrounding the cropland
areas, which may be related to the dry climate that can maintain $NH_3$ in the gaseous
state for a longer time, providing climate conditions for the long distance transmission
of $NH_3$. Recent national measurement work (Pan et al., 2018) also revealed high
surface $NH_3$ concentrations in northwestern Xinjiang, confirming the rationality of the
IASI-derived estimates.

In the US, the overall mean surface $NH_3$ concentrations were 1.52 µg N $m^{-3}$ $y^{-1}$, with
the range of 0.14-10.60 µg N $m^{-3}$. The surface $NH_3$ hotspots were generally
distributed in the croplands in the central and eastern US (such as Ohio, Illinois, South
Dakota, Nebraska, Kansas, Minnesota and North Dakota), as well as in some small
areas in western coastal regions (such as California and Washington). In particular, the
San Joaquin Valley (SJV) in California (an agricultural land) had the highest surface
$NH_3$ concentrations greater than 4 µg N $m^{-3}$. The $NH_3$ source in SJV was from
livestock and mineral N fertilizer, which accounted for 74% and 16% of total $NH_3$
emissions, respectively (Simon et al., 2008). Except the SJV in California, the annual
surface $NH_3$ concentrations in the croplands were mostly within the range of 1-3 µg N
$m^{-3}$, which were much lower than those in eastern China (mostly within the range of
4-10 µg N $m^{-3}$). Compared with the spatial distribution of N fertilization, the hotspots
of surface $NH_3$ concentration can basically reflect the distribution of high N
fertilization ($R^2$=0.30, p=0.000 in **Fig 4 and Fig. 5**).

In Europe, the overall mean surface $NH_3$ concentrations were 1.8 µg N $m^{-3}$, with the





range of 0.04-9.49 $\mu g$ N $m^{-3}$. High surface $NH_3$ concentrations were distributed
widespread in the croplands, especially in the western regions with values greater than
4 $\mu g$ N $m^{-3}$, such as Northern Italy (Milan and its surrounding areas), Switzerland,
central and southern Germany, Eastern France (Paris and its surrounding areas) and
Poland. Overall, there was also a significant linear correlation between surface $NH_3$
concentration and N fertilization ($R^2$=0.17, p=0.000) in Europe, reflecting the
importance of N fertilization on surface $NH_3$ concentration.

$NH_3$ is the most abundant alkaline gas in the atmosphere, and has implications to
neutralize acidic species (such as $H_2SO_4$ and $HNO_3$) to form ammonium salts (such as
$(NH_4)_2SO_4$ and $NH_4NO_3$). Ammonium salts are the important inorganic N
components in PM2.5, which can reduce regional visibility and contribute to human
disease burden (Van et al., 2015;Yu et al., 2007). Comparing surface $NH_3$
concentrations with PM2.5 can benefit the understanding of the sources and the
mixture of air pollution. The spatial distribution of satellite-derived $PM_{2.5}$ (dust and
sea-salt removed) in 2014 (**Fig. S7**) gained from a previous study (Van et al., 2016)
was compared with the satellite-derived surface $NH_3$ concentrations in 2014. On the
other hand, $NO_2$ is also an important precursor of nitrate salts in $PM_{2.5}$. We also
included the satellite-derived surface $NO_2$ concentrations (**Fig. S7**) from a previous
study (Geddes et al., 2016) to compare with surface $NH_3$ and $PM_{2.5}$ concentrations.

The hotspots of surface $NH_3$ concentrations were highly linked with the hotspots of
$PM_{2.5}$. The most severe pollution occurred in the eastern China with annual average
$PM_{2.5}$ exceeding 50 $\mu g$ $m^{-3}$ (much higher than 35 $\mu g$ $m^{-3}$ as the level 2 annual $PM_{2.5}$
standard set by World Health Organization Air Quality Interim Target-1), and annual
average surface $NH_3$ and $NO_2$ concentrations greater than 8 $\mu g$ N $m^{-3}$ and 4 $\mu g$ N $m^{-3}$,
respectively. A previous study (Xu et al., 2017) reported that the secondary inorganic
aerosols of $NH_4^+$ and $NO_3^-$ can account for 65% of $PM_{2.5}$ based on the measurements
in three sites in Beijing. $NH_3$ and $NO_2$ are the most important precursors of nitrate
salts and ammonium salts, and certainly contribute to the severe pollution in the





eastern China. The second severe pollution occurred in the northern India with annual
average $PM_{2.5}$ and surface $NH_3$ concentrations exceeding 40 μg m$^{-3}$ and 4 μg N m$^{-3}$
respectively (surface $NO_2$ concentrations less than 1 μg N m$^{-3}$). The major source of
$NH_3$ in northern India was from agricultural activities and livestock waste
management (Warner et al., 2016). The hotspots of surface $NH_3$ concentrations in the
central and eastern US were highly related to the hotspots of $PM_{2.5}$. The annual
average $PM_{2.5}$ is less than 10 μg m$^{-3}$ (the first level set by World Health Organization)
in the most areas of the US, and only small areas had $PM_{2.5}$ greater than 10 μg m$^{-3}$.
Similarly, in western Europe, the hotspots of high surface $NH_3$ and $NO_2$
concentrations (greater than 3 μg N m$^{-3}$) were consistent with the hotspots of $PM_{2.5}$
(greater than 20 μg m$^{-3}$).

**Seasonal variations of satellite-derived surface $NH_3$ concentrations**

To investigate the seasonal variations of surface $NH_3$ concentrations, we took the
monthly surface $NH_3$ concentrations in 2014 as a case study (**Fig. 6**), and analyzed the
seasonal surface $NH_3$ concentrations in hotspot regions including East China (ECH),
Sichuan and Chongqing (SCH), Guangdong (GD), Northeast India (NEI), East US
(EUS) and West Europe (WEU) (**Fig. 7**).
Seasonal mean IASI-derived surface $NH_3$ concentrations vary by more than 2 orders
of magnitude in hotspot regions, such as the eastern China and eastern US. In China,
high surface $NH_3$ concentrations occurred in spring (March, April and May) and
summer (June, July and August) in East China (ECH), Sichuan and Chongqing (SCH),
Guangdong (GD). This may be due to two major reasons. First, the timing of the
mineral N fertilizer or manure application occurred in summer or spring in the
croplands (Paulot et al., 2014). A previous study (Huang et al., 2012) also suggested a
summer peak in $NH_3$ emissions in China, which was consistent with the summer peak
in surface $NH_3$ concentrations. Second, the temperature in warm months is highest in
one year, which favors the volatilization of ammonium ($NH_4^+ + OH^- \rightarrow NH_3 + H_2O$). In



the eastern US (EUS), high surface NH$_3$ concentrations appeared in warm months
(from March to August, **Fig. 7**) with the maximum in May due to higher temperature
and emissions in vast croplands, where the agricultural mineral N fertilizers dominate
the NH$_3$ emissions. A previous study also implied a spring peak in NH$_3$ emissions in
the eastern US (Gilliland et al., 2006). Since the spatial patterns of high surface NH$_3$
concentrations are highly linked with the spatial distributions of croplands, seasonal
surface NH$_3$ concentrations mainly reflects the timing of N fertilizers in the croplands.
In western Europe, surface NH$_3$ concentrations is low in January and February, rising
in March and reaching its maximum, keeping high from March to June, then declining
from June to December (**Fig. 7**). High NH$_3$ concentrations appeared from March to
June, mainly affected by higher temperature and frequent N fertilization (Van Damme
et al., 2014b;Paulot et al., 2014;Van Damme et al., 2015;Whitburn et al., 2015).

To identify the major regions of biomass burning that may affect the spatial
distribution of surface NH$_3$ concentrations, we used the fire products from the
moderate resolution imaging spectroradiometer (MODIS) on board the NASA Aqua
and Terra. The MODIS climate modeling grid (CMG) global monthly fire location
product (level 2, collection 6) developed by the University of Maryland included
geographic location of fire, raw count of fire pixels and mean fire radiative power
(Giglio et al., 2015). We used the Aqua and Terra CMG fire products on a monthly
scale at a spatial resolution of 0.5 °latitude × 0.5 °longitude in 2014, and the fire pixel
counts were used to identify the hotspot regions of biomass burning. In the major
hotspots with frequent fires (mostly in the southern hemisphere), the biomass burning
controlled the seasonal surface NH$_3$ concentrations (**Fig. S8 and Fig. S9**), such as, for
instance, Africa north of equator, Africa south of equator and central South America.
Apart from the large areas with frequent fires in the southern hemisphere, we also
demonstrated the relationship of biomass burning and surface NH$_3$ concentrations in
China, US and Europe, and selected six typical regions in China (CH1 and CH2), US
(US1 and US2) and Europe (EU1 and EU2) (**Fig. 8**) to analyze the monthly variations
of fire counts and surface NH$_3$ concentrations.




In China, the first region (CH1) covers the major cropland areas in northern China
including Shandong, Henan and and northern Jiangsu Provinces. The fires counts
were mainly from the crop straw burning, which contributes large to surface $NH_3$
concentrations. Both surface $NH_3$ concentrations and fire counts were found in June
likely related to the crop straw burning in this agricultural regions. Notably, despite a
decline in fire counts in July, the surface $NH_3$ concentrations in July did not decrease,
probably due to mineral N fertilization for new planted crops (CH1 is typical for
spring and summer corn rotations) as well as the high temperature favoring $NH_3$
volatilization in July. The second region (CH2) is typical for the rice cultivation area
in the southern China, where the rice was normally planted in June or July with
frequent mineral N fertilization. Thus, the highest surface $NH_3$ concentrations
occurred in June and July. This region is also typical for the winter wheat and summer
rice rotations, and the wheat is normally harvested from May to July, which can lead
to frequent fire counts there. Despite the more frequent fires in the second region than
the first region, the surface $NH_3$ concentrations in CH2 were much lower than those in
CH1. This may be due to the wetter climate and more frequent precipitation events in
CH2 than in CH1, resulting in higher scavenging of surface $NH_3$ concentrations into
water.

US1 is a region typical for forest land in the US, and the fire counts are certainly from
the forest fires or anthropogenic biomass burning. The monthly variations of surface
$NH_3$ concentrations were consistent with the monthly variations of MODIS fire counts,
which peaked in August with high temperature. Instead, US2 is a region typical for
mixed agricultural and forest lands, which can be influenced by both potential mineral
N fertilization and anthropogenic biomass burning or forest fires. It is clear to see that
there is a peak in surface $NH_3$ concentrations in October resulting from the biomass
burning because of the same peak in fire counts in October. However, there is also an
apparent peak in surface $NH_3$ concentrations in May, which may result from the
mineral N fertilization in this region. In Europe, the selected two regions of EU1 and



EU2 are mainly covered by crops, vegetables as well as forests. For EU2, the monthly
variations of surface $NH_3$ concentrations were consistent with the monthly variations
of MODIS fire counts, which peaked in August with high temperature, implying that
the biomass burning may control the seasonal surface $NH_3$ concentrations. For EU1,
there were two peaks of surface $NH_3$ concentrations including April and August. The
August peak can be related to the biomass burning because of the high fire counts,
while the April peak may be related to the agricultural fertilizations for the spring
crops.

### Trends in surface $NH_3$ concentrations in China, the US and Europe

Time series of nine years' (2008-2016) IASI-derived surface $NH_3$ concentrations have
been fitted using the linear regression method (Geddes et al., 2016;Richter et al., 2005)
for all grids on the land. The annual trend (the slope of the linear regression model) is
shown in **Fig. 9**. A significant increase rate of surface $NH_3$ concentrations (>0.2 μg N
$m^{-3}$ $y^{-1}$) appeared in eastern China, and a middle positive trend (0.1-0.2 μg N $m^{-3}$ $y^{-1}$)
occurred in northern Xinjiang, corresponding to its frequent agricultural activities for
fertilized crops and dry climate (Warner et al., 2017;Liu et al., 2017b;Xu et al.,
2015;Huang et al., 2012). The large increase in eastern China was consistent with the
results revealed by AIRS $NH_3$ data (Warner et al., 2017). China's $NH_3$ emissions
increased significantly from 2008 to 2015, with an increase rate of 1.9% $y^{-1}$, which
was mainly driven by eastern China (Zhang et al., 2017b). In addition, the increase in
surface $NH_3$ concentrations in eastern China may be also linked with the decreased
$NH_3$ removal due to the decline in acidic gases ($NO_2$ and $SO_2$) (Liu et al., 2017a;Xia
et al., 2016). $NH_3$ can react with nitric acid and sulfuric acid to form ammonia sulfate
and ammonia nitrate aerosols. The reduction of acidic gases leads to the reduction of
$NH_3$ conversion to ammonia salts in the atmosphere, which may lead to the increase
of $NH_3$ in the atmosphere (Liu et al., 2017a;Li et al., 2017b). China's $SO_2$ emissions
decreased by about 60% in 2008-2016, which leaded to a 50% decrease in surface
$SO_2$ concentrations simulated by WRF model, and then resulted in a 30% increase in



surface NH$_3$ concentrations (Liu et al., 2018).

In the US, the NH$_3$ increase was found in agricultural regions in middle and eastern
regions with an annual increase rate of lower than 0.10 μg N m$^{-3}$ y$^{-1}$, which was
consistent with the results of AIRS NH$_3$ data for a longer time period (2003-2016)
(Warner et al., 2017), while we concerned the timespan of 2008-2016 from IASI
retrievals. Based on the simulation data of CMAQ model, it is also found that NH$_3$
increased significantly in the eastern US from 1990 to 2010, which is inconsistent
with the significant downward trend of NO$_x$ emissions (Zhang et al., 2018). This NH$_3$
increase in eastern US is likely due to the lack of NH$_3$ emission control policy as well
as the decreased NH$_3$ removal due to the decline in acidic gases (NO$_2$ and SO$_2$)
(Warner et al., 2017;Li et al., 2016). As NH$_3$ is an uncontrolled gas in the US, NH$_3$
emissions have continuously increased since 1990, and by 2003 NH$_3$ emissions had
begun to dominate the inorganic N emissions (NO$_x$ plus NH$_3$) (Zhang et al., 2018).
For the western Europe, the trend was close to 0 in most regions although we can
observe the NH$_3$ increase in many points with small positive trend of lower than 0.1
μg N m$^{-3}$ y$^{-1}$. Compared with the trend of surface NH$_3$ concentrations in China and the
US, the change of surface NH$_3$ concentrations in western Europe is more stable,
which may be related to the mature NH$_3$ reduction policies and measures in Europe.
Since 1990, Europe has implemented a series of agricultural NH$_3$ emission reduction
measures, and NH$_3$ emissions decreased by about 29% between 1990 and 2009
(Tørseth et al., 2012). For example, due to serious N eutrophication, the Netherlands
has taken measures to reduce NH$_3$ emissions by nearly two times in the past 20 years,
while maintaining a high level of food production (Dentener et al., 2006). The N
fertilizer use in Europe has decreased widespread according to the data from the
World Bank (http://data.worldbank.org/indicator/AG.CON.FERT.ZS) with an annual
decrease of -8.84~-17.7 kg ha$^{-1}$ y$^{-1}$ in fertilizer use in Europe (Warner et al., 2017).





## Conclusions

The IASI-derived global surface $NH_3$ concentrations during 2008-2016 were inferred based on IASI $NH_3$ column measurements as well as $NH_3$ vertical profiles from the GEOS-Chem in this study. Global $NH_3$ vertical profiles on the land from the GEOS-Chem can be well modelled by the Gaussian function between 60 °N and 55 °S with $R^2$ higher than 0.90. The IASI-derived surface $NH_3$ concentrations were compared to the in situ measurements over China, the US and Europe. One of the major findings is that a relatively high predictive power for annual surface $NH_3$ concentrations was achieved through converting IASI $NH_3$ columns using modelled $NH_3$ vertical profiles, and the validation with the ground-based measurements shows that IASI-derived surface $NH_3$ concentrations had higher accuracy in China than the US and Europe. High surface $NH_3$ concentrations were found in the croplands in China, US and Europe, and surface $NH_3$ concentrations in the croplands in China were approximately double than those in the US and Europe. Seasonal mean IASI-derived surface $NH_3$ concentrations vary by more than 2 orders of magnitude in hotspot regions, such as the eastern China and eastern US. The linear trend analysis shows that a significant positive increase rate of above 0.2 $\mu$g N m$^{-3}$ y$^{-1}$ appeared in the eastern China during 2008-2016, and a middle increase trend (0.1-0.2 $\mu$g N m$^{-3}$ y$^{-1}$) occurred in northern Xinjiang Province. In the US, the $NH_3$ increase was found in agricultural regions in middle and eastern regions with an annual increase rate of lower than 0.10 $\mu$g N m$^{-3}$ y$^{-1}$.

## Author contributions

LL and XZ designed the research; WX and XL's group conducted the field work in China; LL prepared IASI $NH_3$ products; LL and AW conducted model simulations; LL, WX, LZ, XW and ZW performed the data analysis and prepared the figures; LL, AW and XZ wrote the paper, and all coauthors contribute to the revision.





**Acknowledgements**
We acknowledge the free use of IASI NH$_3$ data from the Atmospheric Spectroscopy
Group at Universit élibre de Bruxelles (ULB). We thank Dr. Jeffrrey A. Geddes for the
help of using the GEOS-Chem in this work. This study is supported by the National
Natural Science Foundation of China (No. 41471343, 41425007 and 41101315) and
Doctoral Research Innovation Fund (2016CL07) as well as the Chinese National
Programs on Heavy Air Pollution Mechanisms and Enhanced Prevention Measures
(Project No. 8 in the 2nd Special Program).

**Data availability**
The IASI NH$_3$ satellite datasets are available at: http://iasi.aeris-data.fr/NH3. The
ground-based NH$_3$ mesurements in Chinese Nationwide Nitrogen Deposition
Monitoring Network (NNDMN) can be requested from Prof. Xuejun Liu in China
Agricutural University. The ground-based NH$_3$ measurements from the AMoN-US
can be downloaded from the website: http://nadp.sws.uiuc.edu/AMoN/. The
ground-based NH$_3$ measurements from the EMEP network can be gained from
https://www.nilu.no/projects/ccc/emepdata.html. The IASI-derived surface NH$_3$ used
in this study are available from the corresponding author upon request.

**Notes**
The authors declare that they have no conflict of interest.

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




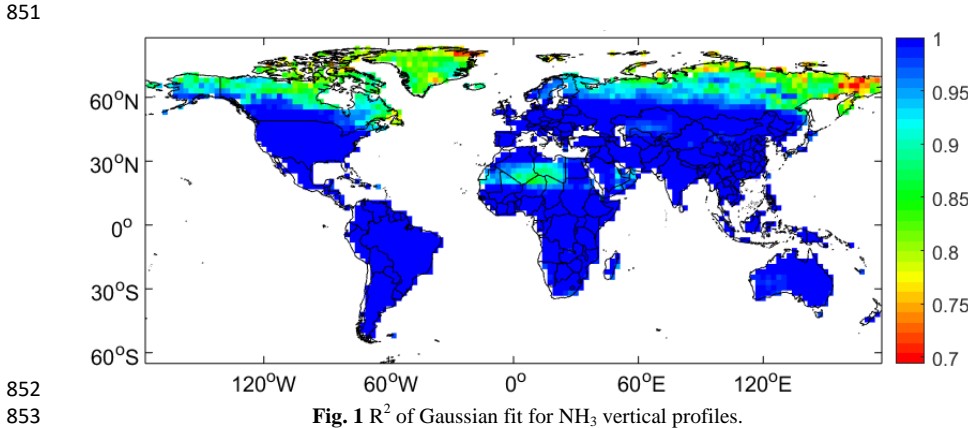

**Fig. 1** $R^2$ of Gaussian fit for $NH_3$ vertical profiles.




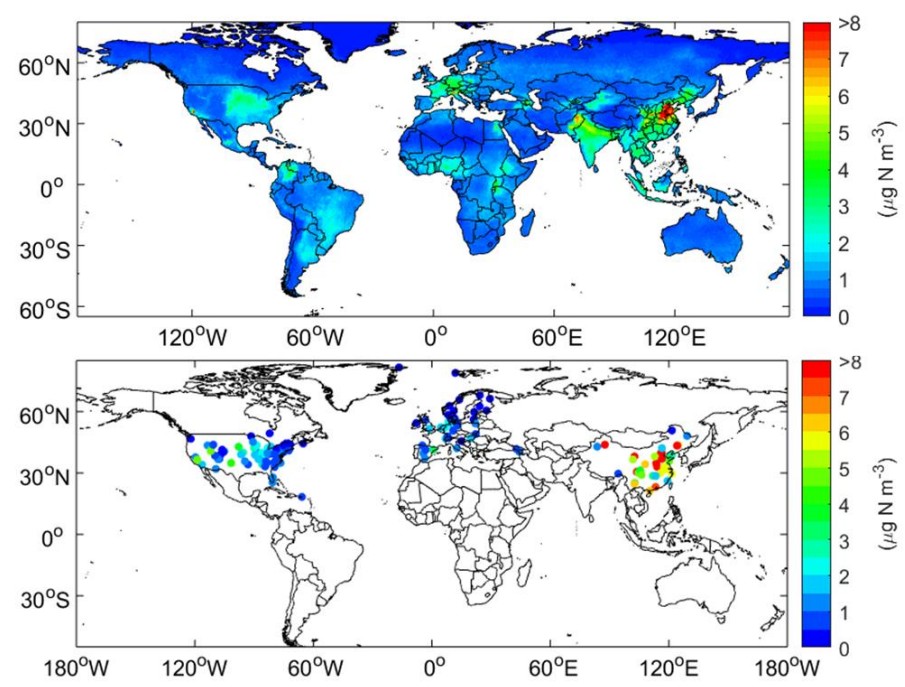

**Fig. 2** Spatial distribution of satellite-derived and measured surface NH$_3$ concentrations in 2014.



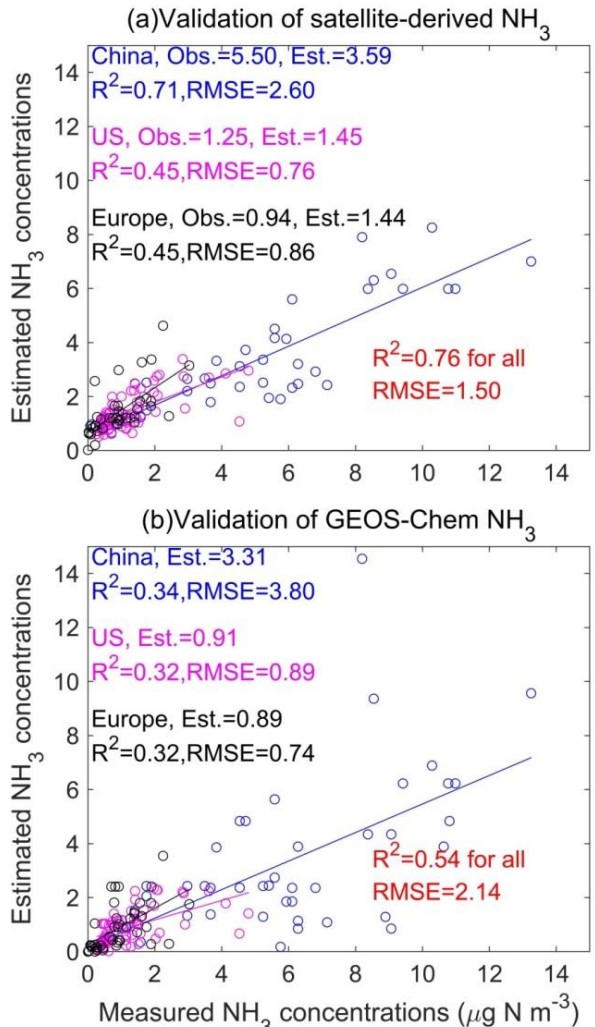


**Fig. 3** Comparison of satellite-derived and GEOS-Chem modelled surface NH$_3$ concentrations with

860          measured concentrations in China, US and Europe.





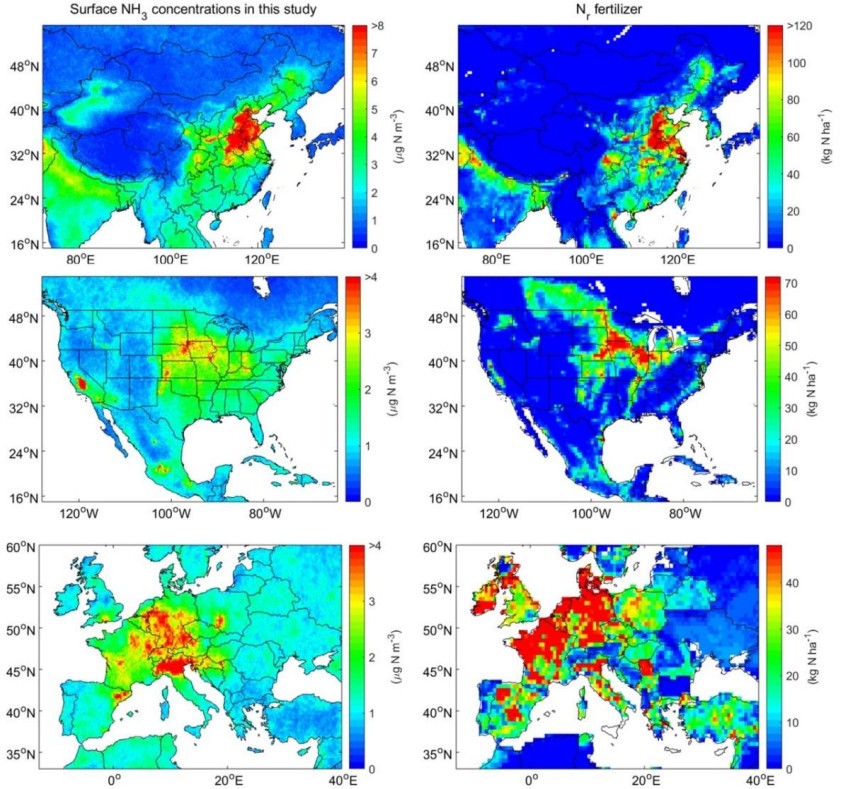


**Fig. 4** Spatial distribution of IASI-derived surface NH$_3$ concentrations and N fertilizer in China, Europe
and US.




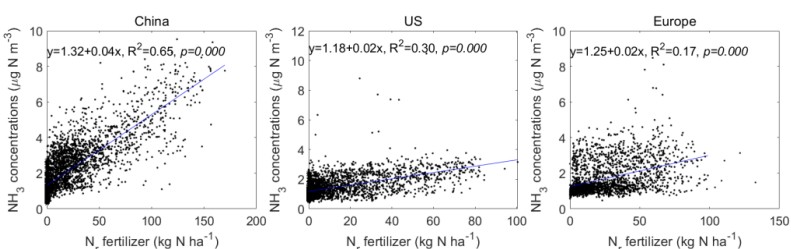


**Fig. 5** Comparison of satellite-derived surface $NH_3$ concentrations and N fertilizer amounts in China,
US and Europe. The spatial resolution of satellite-derived surface $NH_3$ concentrations and N fertilizer is
$0.25^o$ and $0.5^o$, respectively. We firstly resampled the satellite-derived surface $NH_3$ concentrations to
$0.5^o$ grids, and then compared it with N fertilizer data by each grid cell. We obtained the N fertilizer
data produced from McGill University (Potter et al., 2010).




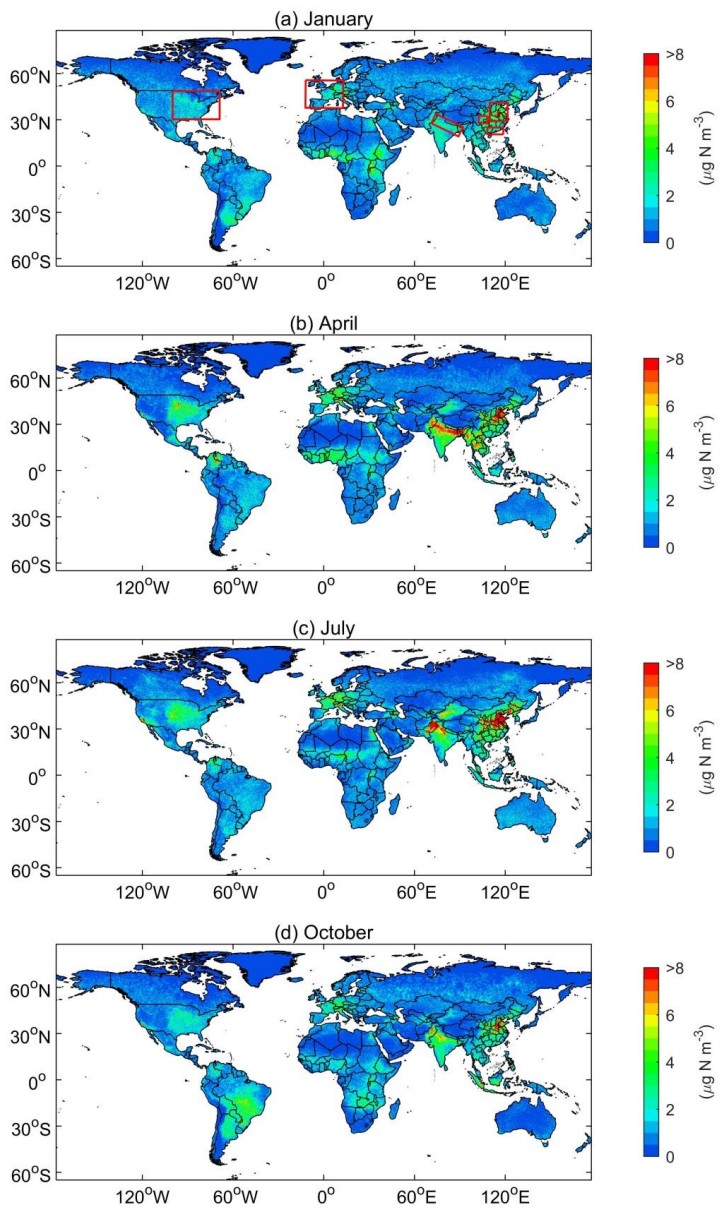


873       **Fig. 6** Global surface NH₃ concentrations in January, April, July and October in 2014. The red

874       rectangular regions include East China (ECH), Sichuan and Chongqing (SCH), Guangdong (GD),

875       Northeast India (NEI), East US (EUS) and West Europe (WEU).



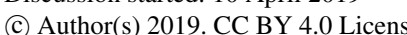


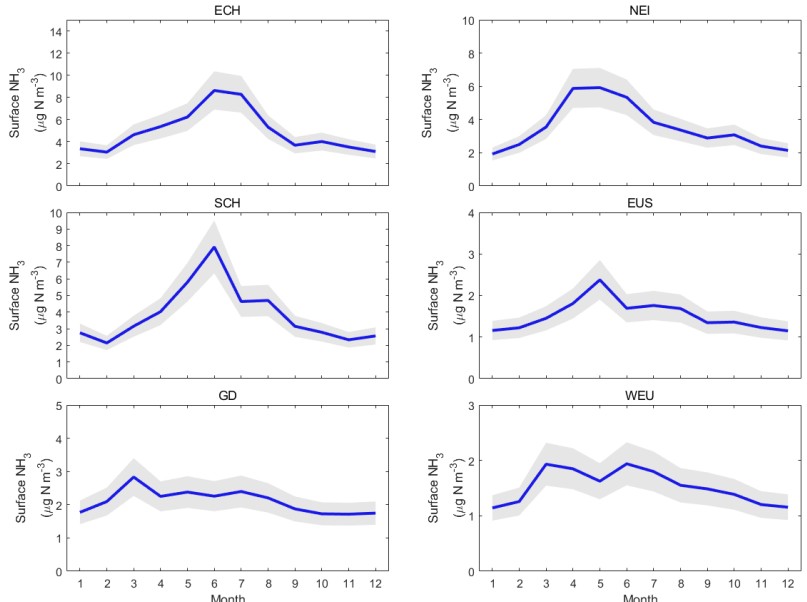

**Fig. 7** Monthly variations of surface NH$_3$ concentrations in hotspot regions including East China (ECH), Sichuan and Chongqing (SCH), Guangdong (GD), Northeast India (NEI), East US (EUS) and West Europe (WEU).




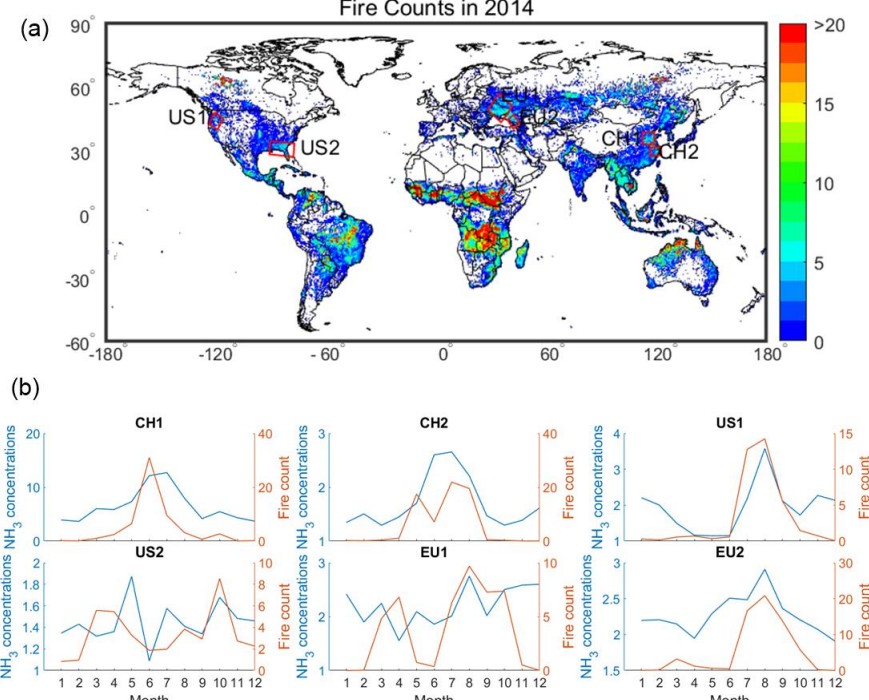


**Fig. 8**. Total raw fire count from the MODIS in 2014 (a), and monthly variations of fire counts and
surface NH$_3$ concentrations in biomass burning regions in China, the US and Europe (b).






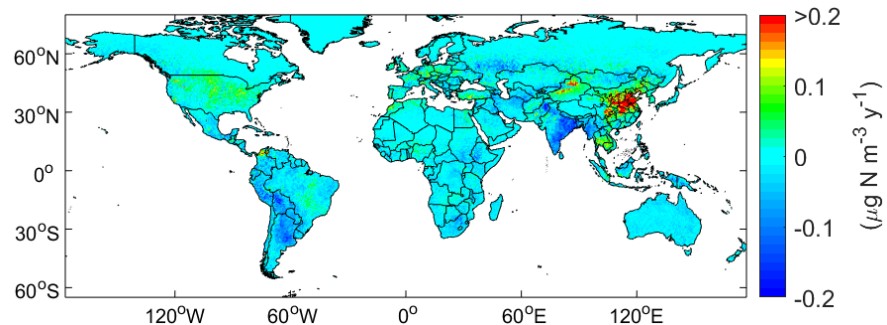

**Fig. 9** Trends of IASI-derived surface NH$_3$ concentrations between 2008 and 2016. A linear regresion
was perfomed to calculate the trends. The significance value (p) and R$^2$ for the trends can be found in
**Fig. S10**.