# Peer review of "Estimating global surface ammonia concentrations inferred"

_Atmospheric Chemistry and Physics, 2019_

## Referee Comment (RC1) · Anonymous Referee #1 · 3 May 2019

This manuscript estimated global surface NH3 concentration based on satellite retrievals and the tempeoral variation of NH3 concentation was also presented. The study was well designed and the results are also important for evaluate NH3 pollution in the world. The major comments to the manuscript are as follows.

1 Lines 278-280, for comparing the satellite-derived and measured surface NH3 concentrations, are there any criterions to choose the sites whith measured surface NH3 concentrations? This is because satellite-derived surface NH3 concentration in a grid (0.25° latitude × 0.25° longitude) is a reflection of the averaged NH3 concentration in this grid area, but the NH3 concentration measured at a site may only represent a limited area. For a grid with diffrent sources of NH3 (e.g., cropland, animal house or feedlot), the NH3 concentration in this grid may have large spatial heterogeneity, then

[Figure]

how to find a site with the surface NH3 concentration to represent a grid area?

2 Lines 282-284, for comaparing NH3 concentrations with different methods, the information on how many measuring sites, and where the sites located should be given for each country or region.

3 Lines 284-286, as mentioned in comment 1, the spatial heterogeneity of NH3 concentration in a grid and the measuring sites location may also cause the differences between satellite-derived and ground-based NH3 concentration. Thus, this discussion should be added here. Besides, the detection limit and precision for deriving NH3 concentraion using the satellite should be given.

4 Lines 318-320, More details of the location of NH3 hotspots should be given. In China, where is the eastern China? It seems that there were also NH3 hotspots in Shannxi,Shanxi, Gusu and Hubei provinces, and there were no hotspot (> 8 ug N m-3)in Xinjiang province in Fig. 4?

5. Lines 321-324, in fact, more than half the NH3 emissions in China is caused by animal production. The higher NH3 concentration in eastern China can also be caused by animal production. More discussion and supporting data should be provided to strenthen the contribution of animal production on NH3 concentration. This is also true for US and Europe.

6. Lines 398-402, are there any differences for the seasonal variation of NH3 concentration in different regions in the world?

7. Lines 486- 488, which sector (crop or animal production) did cause the increase of NH3 emissions in China in 2008-2015?

---

## Referee Comment (RC2) · Anonymous Referee #2 · 13 May 2019

This is a useful, but also rather coarse and preliminary, study using IIASI satellite retrievals to provide information about surface NH3 concentrations and trends. The study uses coarse resolution GEOS-Chem vertical profile information with column information retrieved from IIASI. Overall there is convincing correspondence of the IIASI retrievals with surface concentrations- although more robust statistical analysis beyond showing correlation and bias is precluding understanding how good (or bad) the correspondence really is. How many observations were within a factor of two of the satellite derived values, is there a difference between rural, semi-urban, urban values, etc.

One of the issues that have always remained a bit of mystery to me with regard to the satellite retrievals is the importance of using a-priori profile information for the retrievals, given the lack of sensitivity close to the surface and possibly also saturation effects

where concentrations are high. This paper adds another question mark: when on the one hand the retrieval uses profile information, and in a second step the method uses the same model to calculate surface information is there a potential for using twice the same information? Also unclear to me how the vertical resolution of GEOS-CHEM can resolve the strong vertical gradients that are likely to exist in source regions. The authors should indicate 1) the vertical structure of the model 2) the measurement characteristics of the surface observation (including height), 3) how this information is used to calculate surface concentrations. Even if the authors will not resolve all issues, a thorough discussion and way-forward discussion are needed about this.

For these reasons I recommend major revisions to his paper. I would ask the authors to already put on-line the vertical model information- for use by other readers.

Detailed comments.

l. 24 Not only for dry deposition, also for modelling of the formation of ammonium nitrate.

l. 32: I am wondering what a high correlation of satellite/surface obs taken all regions together is really telling? I think it mostly depends on getting the levels of the 'high' concentrations correct, which is confirmed by the numbers given later. Consider removing this statement

l. 65. They have not been established to measure NH3 by itself, but as one of the parameters of a larger range of pollutants.

l. 110 -116. It seems a selling point to suggest that vertical profile information for NO2 and SO2 have been useful for modelling, and therefore also for NH3. The issues is more complex for NH3, with sources almost entirely close to the surface, and a complex mix of source and sinks, which will make the model profile more dependent on the mixing characteristics of the CTM. For NO2 and SO2 the sources and locations are better known, most of SO2 is nowadays emitted also well above the surface, which

makes interference with the dry deposition process less sensitivity to errors.

l. 134: GEOSCHEM vertical profiles were used in the retrieval and subsequently used to derive surface concentrations=>explain

l. 147. Please summarize some characteristics. E.g. error statistics, characteristic of sites (urban, rural, vicinity to direct sources) height of observations that we need to know to understand the ability of a course resolution model to represent those sites. For all 3 networks!

l.153 Need to explain why temporal averaging to 2 weeks is done.

l. 157 measured systems=>measurement systems

l. 164 Please give the characteristic heights of the layers in the boundary (mixing) layer. And how were emissions distributed?

l. 160 Any spin-up considered?

l. 174: how does this compare to the widely used HTAP2 emissions for 2010? What changes to EDGAR result from substituting with regional inventories. Any seasonality applied and what would be the reference for it?

l. 186 some clarification is need as regards the use of 'local' times, which I think are not considering any shift in legal times (i.e. winter/summertime).

l. 206. It is not clear why this more complex fitting procedure is needed. What was the problem, and how is solved by this new fitting. In figure S2 vertical profiles are shown- but I do not have the information to understand if GEOSCHEMs vertical resolution would be able to resolve such profiles.

l. 223 like mentioned earlier, the correlation is merely due to the fact of having very high concentrations in one region, versus low in other regions. If one would compare the low concentration range a factor of 5 or so difference in concentrations belong to a single column value would be found. And maybe in reality even more, depending on a whole

lot of things. l. 234 also this seems to be a rather error sensitive approach. I am mostly concerned about possible co-variance between the intra-day emission variations and the limited sampling at 9 O'clock. Are there observations that can be used to explore this issue?

L 242 Something wrong with reference E et al. And Van et al. And maybe other references as well.

l. 241-249 I am confused about the Figure S1 (with typical maxima of about 20-40 meter) and the statement that models and aircraft measurements can be used to verify them. I think that high towers like the one at Cabouw (NL), and observations at several levels, are probably the more reliable verification, but unfortunately there are not many of these.

L 250 I have no idea what heights the first and fifth layer are corresponding to.

l. 284 It is true that NH3 can be more accurately be retrieved in one region than another depending on the thermal contrast. But it is not clear to me why this would be so much better in China than e.g. in the US? I guess it is also just a matter of detection limits? It could also be related to more reliable simulation of mixing, depending on sufficient observational input into the parent weather model.

l. 296 the fixed profiles=>fixed profiles (language)

l. 304-312 I believe that there can be a large difference between 40-60 meters, but as the authors explain this is all in the same geos-chem layer. I fail to understand how this information is then used to interpret geos-chem profiles.

l. 323 this is mostly a confirmation that crop mask used by the regional and global emission inventories correspond to the MODIS one. And that the fertilizers used in those countries indeed end up on these fields.

l. 355 According to inventories, in Europe about 20 % of NH3 emissions is related to the use of mineral fertilizer, and 80 % to manure managements. So it would be more

relevant to determine the correspondence of those emissions (mineral is often used to top up what wasn't provided by manure).

364 Van et al. Please check this reference. It seems to be a problem of you reference manager.

370 It is probably opportune to refer to the paper by Pozzer et al, a 1x1 model study that has more extensively studied the role of NH3 emission for aerosol. It is also important to critically assess ammonium nitrate measurements, which are notoriously difficult at higher concentrations. Would it be an option to use the model to estimate (equilibrium) ammonium sulfate and nitrate concentrations associated with the 'retrieved' surface ammonia?

421-430 Biomass burning can be an important source of NH3, especially in the smoldering phase. Therefore, I have some doubts that active fire products are the best proxy for Nh3 emission. Did you consider burnt area products instead?

495 leaded=>led?

504: why inconsistent? It could be rather consistent, as you explain in the following sentences.

Supplementary:

Figure S3: Please indicate what heights approximate correspond with the first and fifth layer boundaries.

Figure S4: no idea what heights these layer correspond to.

Figure S5: the figure caption is not self-explaining.

Figure S6: first PM2.5 and then NO2 in caption.

Figure S9: describe upper panel as well.

Figure S10: trends of what (annual concentrations?) and for what period.

[Figure]

---

## Author Comment (AC1) · 2 Aug 2019

**Response to Referee #1**

This manuscript estimated global surface $NH_3$ concentration based on satellite retrievals and the temporal variation of $NH_3$ concentration was also presented. The study was well designed and the results are also important for evaluate $NH_3$ pollution in the world. The major comments to the manuscript are as follows.

The authors appreciate the valuable suggestions given by Referee #1 for improving the overall quality of the manuscript. In this document, we describe how we addressed the reviewer's comments. Detailed responses to each comment are given below (in blue).

1 Lines 278-280, for comparing the satellite-derived and measured surface $NH_3$ concentrations, are there any criterions to choose the sites which measured surface $NH_3$ concentrations? This is because satellite-derived surface $NH_3$ concentration in a grid ($0.25^o$ latitude $\times$ $0.25^o$ longitude) is a reflection of the averaged $NH_3$ concentration in this grid area, but the $NH_3$ concentration measured at a site may only represent a limited area. For a grid with different sources of $NH_3$ (e.g., cropland, animal house or feedlot), the $NH_3$ concentration in this grid may have large spatial heterogeneity, then how to find a site with the surface $NH_3$ concentration to represent a grid area?

A point-to-grid verification strategy is adopted here, i.e. comparing the measurements at the monitoring stations with the grid values of satellite-derived estimates. We have to admit that this is the uncertainty of our analysis for comparing the satellite-derived and measured surface $NH_3$ concentrations since the monitoring site may not be representative of a given grid cell for an average retrieved value. We have added the following text to discuss this potential uncertainty in the section of "Validation of satellite-derived surface $NH_3$ concentrations":

"Notably, we compared the surface $NH_3$ concentrations at the monitoring stations with

the grid values of satellite-derived estimates directly. This point-to-grid verification strategy may cause uncertainty since the monitoring site location may not be representative of a given grid cell for an average retrieved value.".

2 Lines 282-284, for comparing $NH_3$ concentrations with different methods, the information on how many measuring sites, and where the sites located should be given for each country or region.

Thanks very much for this good suggestion. We have added the number of measuring sites in each region by the following text:

"IASI-derived surface $NH_3$ concentrations gained higher consistency with the ground-based measurements in China ($R^2$=0.71 and RMSE=2.6 µg N m$^{-3}$ for 43 sites) than the US ($R^2$=0.45 and RMSE=0.76 µg N m$^{-3}$ for 67 sites) and Europe ($R^2$=0.45 and RMSE=0.86 µg N m$^{-3}$ for 43 sites) at a yearly scale".

The sites locations have been given for each region in Fig. 2 in the manuscript .

3 Lines 284-286, as mentioned in comment 1, the spatial heterogeneity of $NH_3$ concentration in a grid and the measuring sites location may also cause the differences between satellite-derived and ground-based $NH_3$ concentration. Thus, this discussion should be added here.

This concern has been addressed in the response to comment 1. Please refer to it.

Besides, the detection limit and precision for deriving $NH_3$ concentration using the satellite should be given.

Thanks very much for this good suggestion. We have added the following text for clarifications:

"The satellite-derived $NH_3$ has a detection limit of 0.0025 µg N m$^{-3}$ (2.5 ppb) (Graaf et al. 2018; Van Damme et al. 2014).".

4 Lines 318-320, More details of the location of $NH_3$ hotspots should be given. In China, where is the eastern China? It seems that there were also $NH_3$ hotspots in Shannxi, Shanxi, Gansu and Hubei provinces, and there were no hotspot (> 8 ug N m$^{-3}$) in Xinjiang province in Fig. 4?

Thanks very much for this good comment. We have revised the "eastern China" as "eastern China (109-122$^o$ E, 28-41$^o$ N)".

To show more details of the locations of $NH_3$ hotspots, we have revised the descriptions by the following text:

"We found large areas in eastern China (109-122$^o$ E, 28-41$^o$ N), Sichuan Basin, Hubei (including Wuhan, Xiangyang and Yichang), Shaanxi (including Xi'an, Baoji, Hanzhong, Weinan), Gansu (Lanzhou and its surrounding areas), Shanxi (including Yuncheng and Changzhi) and northwestern Xinjiang with surface $NH_3$ concentrations greater than 8 μg N m$^{-3}$ y$^{-1}$.".

5. Lines 321-324, in fact, more than half the $NH_3$ emissions in China is caused by animal production. The higher $NH_3$ concentration in eastern China can also be caused by animal production. More discussion and supporting data should be provided to strengthen the contribution of animal production on $NH_3$ concentration. This is also true for US and Europe.

Thank you very much for this suggestion. We have carefully checked the $NH_3$ emissions. In addition to N fertilization, N manure is another major source of $NH_3$ emissions in China, and the percentage of N manure to $NH_3$ emissions exceeds 50% (Kang et al. 2016). So we have added the N manure into our analysis in Fig. 4 and Fig. 5 and revised related text in the discussion.

[Figure]

**Fig. 4** Spatial distribution of IASI-derived surface NH₃ concentrations, and N fertilizer plus N manure in China, Europe and US.

[Figure]

**Fig. 5** Comparison of satellite-derived surface NH₃ concentrations, and N fertilizer plus N manure in China, US and Europe. The spatial resolution of satellite-derived surface NH₃ concentrations and N fertilizer plus N manure is 0.25° and 0.5°, respectively. We firstly resampled the satellite-derived surface NH₃ concentrations to 0.5° grids, and then compared it with N fertilizer plus N manure by each grid cell. We obtained the N fertilizer and N manure data produced from McGill University (Potter et al. 2010).

6. Lines 398-402, are there any differences for the seasonal variation of NH₃ concentration in different regions?

Yes. We take a case study on the seasonal NH₃ concentration in two hotspots of eastern China and Guangdong. The maximum surface NH₃ concentration in eastern

China occurred in June and July, which coincided with the planting, fertilization time and higher temperature of the main crops in the region (Van Damme et al. 2015). The maximum surface $NH_3$ concentration appeared in March in Guangdong, which was also closely related to crop fertilization in these areas (Shen et al. 2009; Van Damme et al. 2014). We have added the following text for clarifications:

"Notably, there is a difference in the seasonal variations of surface $NH_3$ concentrations between ECH (peaking in June and July) and GD (peaking in March), which was likely related to different crop planting, N fertilization time as well as meteorological factors (Shen et al. 2009; Van Damme et al. 2014; Van Damme et al. 2015).".

7. Lines 486- 488, which sector (crop or animal production) did cause the increase of $NH_3$ emissions in China in 2008-2015?

We have added the following text to explore the potential reasons:

"The increase of surface $NH_3$ concentrations in eastern China was consistent with the trend of $NH_3$ emission estimates by a recent study (Zhang et al. 2017). Approximately 85% of the inter-annual variations was due to the changes of human activities, and the remaining 15% resulted from air temperature changes. Agricultural activities is the main drive of $NH_3$ emission increase, of which 43.1% and 36.4% were contributed by livestock manure and fertilizer application (Zhang et al. 2017).".

**Reference**

Graaf, S.C.v.d., Dammers, E., Schaap, M., & Erisman, J.W. (2018). How are $NH_3$ dry deposition estimates affected by combining the LOTOS-EUROS model with IASI-NH3 satellite observations? *Atmospheric Chemistry and Physics, 18*, 13173-13196

Kang, Y., Liu, M., Song, Y., Huang, X., Yao, H., Cai, X., Zhang, H., Kang, L., Liu, X., Yan, X., He, H., Zhang, Q., Shao, M., & Zhu, T. (2016). High-resolution ammonia emissions inventories in China from 1980 to 2012. *Atmospheric Chemistry and Physics, 16*, 2043-2058

Potter, P., Ramankutty, N., Bennett, E.M., & Donner, S.D. (2010). Characterizing the Spatial Patterns of Global Fertilizer Application and Manure Production. *Earth Interactions, 14*, 1-22

Shen, J.L., Tang, A.H., Liu, X.J., Fangmeier, A., Goulding, K.T.W., & Zhang, F.S. (2009). High concentrations and dry deposition of reactive nitrogen species at two sites in the North China Plain. *Environmental Pollution, 157*, 3106-3113

Van Damme, M., Clarisse, L., Dammers, E., Liu, X., Nowak, J., Clerbaux, C., Flechard, C., Galy-Lacaux, C., Xu, W., & Neuman, J. (2014). Towards validation of ammonia ($NH_3$) measurements from the IASI satellite. *Atmospheric Measurement Techniques, 7*, 12125-12172

Van Damme, M., Erisman, J.W., Clarisse, L., Dammers, E., Whitburn, S., Clerbaux, C., Dolman, A.J., & Coheur, P.F. (2015). Worldwide spatiotemporal atmospheric ammonia ($NH_3$) columns variability revealed by satellite. *Geophysical Research Letters, 42*, 8660-8668

Zhang, X., Wu, Y., Liu, X., Reis, S., Jin, J., Dragosits, U., Van Damme, M., Clarisse, L., Whitburn, S., Coheur, P.-F., & Gu, B. (2017). Ammonia Emissions May Be Substantially Underestimated in China. *Environmental science & technology, 51*, 12089-12096

---

## Author Comment (AC2) · 2 Aug 2019

**Response to Referee #2**

We thank the reviewer very much for the detailed and valuable comments. We believe that addressing the issues raised by the reviewer will considerably improve the quality of our manuscript. Please see our response to each comment below (in blue).

**Received and published: 13 May 2019**

This is a useful, but also rather coarse and preliminary, study using IASI satellite retrievals to provide information about surface  $NH_3$  concentrations and trends. The study uses coarse resolution GEOS-Chem vertical profile information with column information retrieved from IASI. Overall there is convincing correspondence of the IASI retrievals with surface concentrations, although more robust statistical analysis beyond showing correlation and bias is precluding understanding how good (or bad) the correspondence really is. How many observations were within a factor of two of the satellite derived values, is there a difference between rural, semi-urban, urban values, etc.

Thanks very much for this comment. We have added the following text for clarifications in the section of "Validation of satellite-derived surface NH3 concentrations":

"Overall, 72.85% of observations (including China, the US and Europe) were within a factor of two of the satellite-derived surface NH3 concentrations. In China, there is approximately 71.43% and 77.27% of observations were within a factor of two of the satellite-derived surface NH3 concentrations in urban and rural land uses, respectively. There is no big difference in the accuracy of satellite-derived surface NH3 concentrations between urban and rural land uses. In the US, the monitoring sites were generally distributed at rural sites (http://www.radiello.com) (Li et al. 2016), and, in Europe, there is no information to indicate the land use of each site (https://projects.nilu.no//ccc/) (T ørseth et al. 2012)."

One of the issues that have always remained a bit of mystery to me with regard to the satellite retrievals is the importance of using a-priori profile information for the

retrievals, given the lack of sensitivity close to the surface and possibly also saturation effects where concentrations are high. This paper adds another question mark: when on the one hand the retrieval uses profile information, and in a second step the method uses the same model to calculate surface information is there a potential for using twice the same information?

We have added the following text for more clarifications (the purposes of using the profiles twice were different) in Sect. "IASI NH3 measurements":

"The ANNI-NH3-v2.2R-I datasets used the ANNI algorithm and took account of the influence of the NH3 vertical profiles, pressure, humidity and temperature profiles, which was to make the columns accurate. There is no information on NH3 vertical profiles in the ANNI-NH3-v2.2R-I datasets. The NH3 vertical profiles used in this paper was to convert the columns to surface concentrations and to make the surface NH3 estimates accurate."

Also unclear to me how the vertical resolution of GEOSCHEM can resolve the strong vertical gradients that are likely to exist in source regions. The authors should indicate 1) the vertical structure of the model, 2) the measurement characteristics of the surface observation (including height), 3) how this information is used to calculate surface concentrations. Even if the authors will not resolve all issues, a thorough discussion and way-forward discussion are needed about this.

Thanks very much for this good comment. We have added the following text for clarifications:

"The IASI NH3 data we gained are column data, and there is no information on the vertical information. To convert this columns to surface concentrations, we used the widely used modelled vertical profiles from GEOS-Chem. The GEOS-Chem outputs include 47 layers, which are not continuous in the vertical direction. To gain the continuous vertical NH3 profile, we used the Gaussian function to fit the 47 layers' NH3 concentrations. The main advantage to simulate the vertical profiles is that the NH3 concentration at any height indicated by satellite can be obtained. On the other hand, the simulated profile function has a general rule, which can convert the columns indicated by satellite to surface concentration simply and quickly."

The GEOS-Chem can simulate the  $NH_3$  vertical profiles at 47 layers, and can simulate  $NH_3$  concentrations at each layer (from approximately 50 m to 20000 m). Most of all the sites in China, US and Europe were set a height of 1-50 m above the ground (Li et al. 2016; Xu et al. 2015). Please note that the height that we mean here is the height to the ground rather than the height above sea level.

For these reasons I recommend major revisions to his paper.

I would ask the authors to already put on-line the vertical model information for use by other readers.

Thanks very much for this good suggestion. We added the  $NH_3$  vertical model information (Matlab code) as the supporting materials in this revision.

**Detailed comments**

1. 24 Not only for dry deposition, also for modelling of the formation of ammonium nitrate.

**We have added it as suggested.**

1. 32: I am wondering what a high correlation of satellite/surface obs taken all regions together is really telling? I think it mostly depends on getting the levels of the 'high' concentrations correct, which is confirmed by the numbers given later. Consider removing this statement.

This indicates the overall accuracy assessment of satellite-based estimates compared with all observations, which include not only "high" concentrations but also "low" concentrations.

1. 65. They have not been established to measure  $NH_3$  by itself, but as one of the parameters of a larger range of pollutants.

We have changed it as suggested.

1. 110 -116. It seems a selling point to suggest that vertical profile information for  $NO_2$  and  $SO_2$  have been useful for modelling, and therefore also for  $NH_3$ . The issues is more complex for  $NH_3$ , with sources almost entirely close to the surface, and a complex mix of source and sinks, which will make the model profile more dependent

on the mixing characteristics of the CTM. For  $NO_2$  and  $SO_2$  the sources and locations are better known, most of  $SO_2$  is nowadays emitted also well above the surface, which makes interference with the dry deposition process less sensitivity to errors.

We agree with you that there are different sources and sinks between NH3, NO2 and SO2. There is no causal relationship of using vertical profile information for NO2, SO2 as well as for NH3. IASI-derived surface NH3 concentrations combining NH3 vertical profiles from CTMs in China and Europe were evaluated previously (Graaf et al. 2018; Liu et al. 2017). Following these studies, the aim of this paper is to determine for the applicability and the assessment of using IASI retrievals and the vertical profile for global surface NH3 concentrations. Our results shows that the satellite-based approach achieved a high predictive power for annual surface NH3 concentrations compared with the measurements of all sites in China, US and Europe (R2=0.76 and RMSE=1.50 µg N m-3).

1. 134: GEOSCHEM vertical profiles were used in the retrieval and subsequently used to derive surface concentrations=>explain

We have added the following text for more clarifications in Sect. "IASI NH3 measurements":

"The ANNI-NH3-v2.2R-I datasets used the ANNI algorithm and took account of the influence of the NH3 vertical profiles, pressure, humidity and temperature profiles, which was just to make the columns accurate. There is no information on NH3 vertical profiles in the ANNI-NH3-v2.2R-I datasets. The NH3 vertical profiles used in this paper was to convert the columns to surface concentrations and to make the surface NH3 estimates accurate."

1. 147. Please summarize some characteristics. E.g. error statistics, characteristic of sites (urban, rural, vicinity to direct sources) height of observations.

We have added the following text for clarifications in the section of "Surface NH3 measurements":

"In China, we used the national measurements from the Chinese Nationwide Nitrogen Deposition Monitoring Network (NNDMN) including 10 urban sites, 22 rural sites, and 11 background sites. The precision for monthly measurements at a site using DELTA systems is as below approximately 10% (Sutton et al. 2001), the correlation between the ALPHA and DELTA measurements was highly significant ( $R^2$ =0.919, p<0.001) (Xu et al. 2015).

Surface NH3 concentrations in the AMoN-US were measured by the radiello diffusive sampler (http://www.radiello.com) as a simple diffusion-type sampler collected every 2 weeks, and these sites were generally distributed at rural sites (Li et al. 2016).

The overall bias of the different instruments in EMEP varied from -30 to 10% for all sites (Bobrutzki et al. 2010). Most of all the sites in China, US and Europe were set a height of 1-50 m above the ground (Li et al. 2016; Puchalski et al. 2011; Xu et al. 2015).".

1. 153 Need to explain why temporal averaging to 2 weeks is done.

We have added the following text for clarification:

"Surface NH3 concentrations in the AMoN-US were measured by the radiello diffusive sampler (http://www.radiello.com) as a simple diffusion-type sampler collected every 2 weeks (Li et al. 2016). We calculated annual surface NH3 concentrations by averaging all the measurements since we compared the measured surface NH3 concentrations with satellite-derived surface NH3 concentrations on a yearly scale.".

1. 157 measured systems=>measurement systems

We have replaced "measured systems" with "measurement systems".

1. 164 Please give the characteristic heights of the layers in the boundary (mixing) layer.

We have added the following text in the caption of Fig. S4 to show more information: "The middle height of 1, 10, 20, 30 and 40 layer was approximately 60, 700, 2000, 6000 and 10000 m, respectively.".

1. 160 Any spin-up considered?

Yes, we have added the following text for clarifications:

"We have done the spin up for 5 months, which well exceed the typical lifetime of atmospheric  $NH_3$  (typically within 24 hours) and aerosol ammonium ions (typically within a week) (Pye et al. 2009)."

1. 174: how does this compare to the widely used HTAP2 emissions for 2010? What changes to EDGAR result from substituting with regional inventories. Any seasonality applied and what would be the reference for it?

HTAP v2.2 is constructed by harmonizing regional emission inventories from USA, Canada, Europe and Asia, and gap-filling the rest of the world by EDGAR v4.3 (Janssens-Maenhout et al., 2015), which is methodologically consistent with our emission configuration. We compared the emissions from EDGAR v4.2 with HTAP v2.2 at 2008, which have differences in global total NH3 within 10% (Crippa et al., 2018). The main difference between the regional inventories and EDGAR is that seasonality of emissions is included in regional inventories,. The seasonality of the regional emissions inventories is embedded as integral part of the inventory except EMEP (Crippa et al. 2018; Janssens-Maenhout et al. 2015; Lenhart and Friedrich 1995). We have added the above explanations in the section of "GEOS-Chem model". 1. 186 some clarification is need as regards the use of 'local' times, which I think are not considering any shift in legal times (i.e. winter/summertime).

We have added the following text for more clarification, and there is no need to consider shift in legal times:

"The local time is the time in a particular region or area expressed with reference to the meridian passing through it.".

1. 206. It is not clear why this more complex fitting procedure is needed. What was the problem, and how is solved by this new fitting. In figure S2 vertical profiles are shown but I do not have the information to understand if GEOSCHEMs vertical resolution would be able to resolve such profiles.

Please refer to the first paragraph in the section of "Estimation of surface  $NH_3$  concentrations. The IASI  $NH_3$  data we gained are column data, and there is no information on the vertical information. To convert the columns to surface concentrations, we used the widely used modelled vertical profiles from GEOS-Chem. The GEOS-Chem outputs include 47 layers, which are not continuous in the vertical direction. To gain the continuous vertical  $NH_3$  profile, we used the Gaussian function to fit the 47 layers'  $NH_3$  concentrations. The main advantage to simulate the vertical

profiles is that the  $NH_3$  concentration at any height indicated by satellite can be obtained. On the other hand, the simulated profile function has a general rule, which can convert the columns indicated by satellite to surface concentration simply and quickly.

1. 223 like mentioned earlier, the correlation is merely due to the fact of having very high concentrations in one region, versus low in other regions. If one would compare the low concentration range a factor of 5 or so difference in concentrations belong to a single column value would be found. And maybe in reality even more, depending on a whole lot of things.

We agree with you that, on a local scale, the relationship of surface  $NH_3$  concentrations and  $NH_3$  columns may be affected by many factors. Here we show the overall accuracy assessment on a global scale between the surface  $NH_3$  concentrations and  $NH_3$  columns based on the GEOS-Chem outputs, which include not only "high" concentrations but also "low" concentrations.

1. 234 also this seems to be a rather error sensitive approach. I am mostly concerned about possible co-variance between the intra-day emission variations and the limited sampling at 9 O'clock. Are there observations that can be used to explore this issue? No, we donot have observations to validate the intra-day emission variations.

L 242 Something wrong with reference E et al. And Van et al. And maybe other references as well.

We have corrected all these references.

1. 241-249 I am confused about the Figure S1 (with typical maxima of about 20-40 meter) and the statement that models and aircraft measurements can be used to verify them. I think that high towers like the one at Cabouw (NL), and observations at several levels, are probably the more reliable verification, but unfortunately there are not many of these.

Fig. S1 just show an example of a possible  $NH_3$  vertical profile, and the typical maxima can be any height such as 20 m, 200 m or other values. In the future we hope to have more aircraft measurements to validate the simulations.

L 250 I have no idea what heights the first and fifth layer are corresponding to.

We have added the following text in the caption of Fig. S3 to show more information: "The middle height of first and fifth layer was approximately 60 m and 340 m, respectively.".

1. 284 It is true that  $NH_3$  can be more accurately be retrieved in one region than another depending on the thermal contrast. But it is not clear to me why this would be so much better in China than e.g. in the US? I guess it is also just a matter of detection limits? It could also be related to more reliable simulation of mixing, depending on sufficient observational input into the parent weather model.

We agree with you that the accuracy of IASI-retrieved surface  $NH_3$  concentrations in different regions is highly linked with the thermal contrast (TC) and the simulation of  $NH_3$  mixing from GEOS-Chem. We have added the following text to discuss the potential reasons.

"The accuracy of IASI-retrieved surface NH3 concentrations in different regions is highly linked with the thermal contrast (TC) and atmosphere NH3 abundance (Whitburn et al. 2016). The lowest uncertainties occurred when high columns and high TC coincide. In case either of them decrease, the uncertainty will gradually increase. In case both the TC and column are low, all sensitivity to NH3 is lost. When high TC and high NH3 columns (high HRI) occurs, the major contribution to the uncertainty results from the thickness of the NH3 layer, the surface temperature as well as the temperature profile (Whitburn et al. 2016). In addition, the simulation of NH3 mixing from GEOS-Chem may have different accuracy in different regions, and thus can cause uncertainty to the different accuracy of IASI-retrieved surface NH3 concentrations in different regions.".

1. 296 the fixed profiles=>fixed profiles (language)

We have corrected it as suggested.

1. 304-312 I believe that there can be a large difference between 40-60 meters, but as the authors explain this is all in the same geos-chem layer. I fail to understand how this information is then used to interpret geos-chem profiles.

We used the equation (2) to fit  $NH_3$  vertical profiles at each grid box by the following equation (Liu et al. 2017):

 $\rho = \sum_{i=1}^{n} \rho_{max,i} e^{-(\frac{Z-Z_{0,i}}{\sigma_i})^2}$  (2)

Once the NH3 vertical profiles were determined at each grid box, we can extrapolate NH3 concentrations at any height from the GEOS-Chem ( $G_{GEOS-Chem}$ ).

Then we can calculate the IASI-derived NH3 concentrations any height using the NH3 vertical profiles and IASI NH3 columns:

$$\overline{G_{IASI_{9-10}}} = \frac{G_{GEOS-Chem}}{\Omega_{GEOS-Chem}} \times \overline{\Omega_{IASI_{9-10}}} \quad (3)$$

where  $\overline{G_{IASI_{9-10}}}$  is the satellite-derived surface NH3 concentrations at a GEOS-Chem grid size at 9-10am;  $\frac{G_{GEOS-Chem}}{\Omega_{GEOS-Chem}}$  is the ratio of surface NH3 concentrations to NH3 columns calculated from GEOS-Chem;  $\overline{\Omega_{IASI_{9-10}}}$  is the average IASI NH3 columns at a GEOS-Chem grid at 9-10am.

**All the information has been described in detail in the method section.**

1. 323 this is mostly a confirmation that crop mask used by the regional and global emission inventories correspond to the MODIS one. And that the fertilizers used in those countries indeed end up on these fields.

**Yes, it is.**

1. 355 According to inventories, in Europe about 20 % of  $NH_3$  emissions is related to the use of mineral fertilizer, and 80 % to manure managements. So it would be more relevant to determine the correspondence of those emissions (mineral is often used to top up what wasn't provided by manure).

Thank you very much for this suggestion. We have carefully checked the  $NH_3$  emissions in Europe. According to Emissions Database for Global Atmospheric Research (EDGAR), in western Europe, manure management accounts for 53% and the share of emissions from agricultural soils for 43% of the ammonia emissions. So we have added the N manure into our analysis in Fig. 4 and Fig. 5 and revised related text in the discussion.

364 Van et al. Please check this reference. It seems to be a problem of your reference manager.

We have checked this reference, and revised it.

370 It is probably opportune to refer to the paper by Pozzer et al, a 1x1 model study that has more extensively studied the role of NH3 emission for aerosol. It is also important to critically assess ammonium nitrate measurements, which are notoriously difficult at higher concentrations. Would it be an option to use the model to estimate (equilibrium) ammonium sulfate and nitrate concentrations associated with the 'retrieved' surface ammonia?

We have carefully read the suggested paper on the the role of  $NH_3$  emission for aerosol (Pozzer et al. 2012). We agree with the reviewer that it is important to assess ammonium nitrate measurements associated with the retrieved surface  $NH_3$ concentrations. However, this has been out of the scope of this paper, and this paper focuses on the estimates of surface  $NH_3$  concentrations inferred from satellite retrievals. It is more appropriate to estimate ammonium sulfate and nitrate concentrations in another paper in the future.

421-430 Biomass burning can be an important source of  $NH_3$ , especially in the smoldering phase. Therefore, I have some doubts that active fire products are the best proxy for  $NH_3$  emission. Did you consider burnt area products instead?

We here compared the monthly variations of surface  $NH_3$  concentrations and biomass burning. The MODIS active fires are considered to be more accurate than the burnt area products on the timing of burning. Please see the temporal intercomparison of burned area products with Active Fire data set (Humber et al. 2019).

activity leaded=>led?

We have replaced "leaded" with "led".

504: why inconsistent? It could be rather consistent, as you explain in the following sentences.

We have added the following text to explain this inconsistency:

"This inconsistency between  $NH_3$  and  $NO_x$  trends in the US was mainly due to different emission control policies. Over the past two decades, due to the implementation of effective regulations and emission reduction measures for  $NO_x$ , the  $NO_x$  emission in the US decreased by nearly 41% between 1990 and 2010 (Hand et al. 2014). However, this  $NH_3$  increase in eastern US is likely due to the lack of  $NH_3$  emission control policy as well as the decreased  $NH_3$  removal due to the decline in acidic gases (NO2 and SO2) (Li et al. 2016; Warner et al. 2017).".

**Supplementary**

Figure S3: Please indicate what heights approximate correspond with the first and fifth layer boundaries.

We have added the following text in the figure caption:

"The middle height of first and fifth layer was approximately 60 m and 340 m, respectively.".

Figure S4: no idea what heights these layer correspond to.

We have added the following text in the figure caption:

"The middle height of 1, 10, 20, 30 and 40 layer was approximately 60, 700, 2000, 6000 and 10000 m, respectively.".

Figure S5: the figure caption is not self-explaining.

We have revised the figure caption by the following text to better describe the figure:

"Difference of surface NH3 concentrations between 40m and 60m.".

Figure S6: first PM2.5 and then NO2 in caption.

We have corrected it in the caption to match it with the figure.

Figure S9: describe upper panel as well.

We have added the following text to describe the panel:

"The upper panel is the annual raw fire counts in 2014.".

Figure S10: trends of what (annual concentrations?) and for what period

Yes, it is annual concentration during 2008-2016. We have changed the original descriptions by the following text:

"...trends of annual surface NH3 concentrations during 2008-2016".

**Reference**

Bobrutzki, K.V., Braban, C.F., Famulari, D., Jones, S.K., Blackall, T., Smith, T.E.L., Blom, M., Coe, H., Gallagher, M., & Ghalaieny, M. (2010). Field inter-comparison of eleven atmospheric ammonia measurement techniques. *Atmospheric Measurement*

**Techniques, 3, 1(2010-01-27), 2, 91-112**

Crippa, M., Guizzardi, D., Muntean, M., Schaaf, E., Dentener, F., van Aardenne, J.A., Monni, S., Doering, U., Olivier, J.G.J., Pagliari, V., & Janssens-Maenhout, G. (2018). Gridded emissions of air pollutants for the period 1970–2012 within EDGAR v4.3.2. *Earth Syst. Sci. Data, 10*, 1987-2013

Graaf, S.C.v.d., Dammers, E., Schaap, M., & Erisman, J.W. (2018). How are NH3 dry deposition estimates affected by combining the LOTOS-EUROS model with IASI-NH3 satellite observations? *Atmospheric Chemistry and Physics*, *18*, 13173-13196

Hand, J.L., Schichtel, B.A., Malm, W.C., Copeland, S., Molenar, J.V., Frank, N., & Pitchford, M. (2014). Widespread reductions in haze across the United States from the early 1990s through 2011. *Atmospheric Environment*, *94*, 671-679

Humber, M.L., Boschetti, L., Giglio, L., & Justice, C.O. (2019). Spatial and temporal intercomparison of four global burned area products. *International Journal of Digital Earth*, *12*, 460-484

Janssens-Maenhout, G., Crippa, M., Guizzardi, D., Dentener, F., Muntean, M., Pouliot, G., Keating, T., Zhang, Q., Kurokawa, J., Wankmüller, R., Denier van der Gon, H., Kuenen, J.J.P., Klimont, Z., Frost, G., Darras, S., Koffi, B., & Li, M. (2015). HTAP\_v2.2: a mosaic of regional and global emission grid maps for 2008 and 2010 to study hemispheric transport of air pollution. *Atmos. Chem. Phys.*, *15*, 11411-11432

Lenhart, L., & Friedrich, R. (1995). European emission data with high temporal and spatial resolution. *Water Air & Soil Pollution*, *85*, 1897-1902

Li, Y., Schichtel, B.A., Walker, J.T., Schwede, D.B., Chen, X., Lehmann, C.M., Puchalski, M.A., Gay, D.A., & Collett, J.L. (2016). Increasing importance of deposition of reduced nitrogen in the United States. *Proceedings of the National Academy of Sciences*, *113*, 5874-5879

Liu, L., Zhang, X., Xu, W., Liu, X., Lu, X., Wang, S., Zhang, W., & Zhao, L. (2017). Ground Ammonia Concentrations over China Derived from Satellite and Atmospheric Transport Modeling. *Remote Sensing*, *9*, 467

Pozzer, A., de Meij, A., Pringle, K.J., Tost, H., Doering, U.M., van Aardenne, J., &

Lelieveld, J. (2012). Distributions and regional budgets of aerosols and their precursors simulated with the EMAC chemistry-climate model. *Atmos. Chem. Phys.*, *12*, 961-987

Puchalski, M.A., Sather, M.E., Walker, J.T., Lehmann, C.M.B., Gay, D.A., Johnson,
M., & Robarge, W.P. (2011). Passive ammonia monitoring in the United States:
comparing three different sampling devices. *Journal of Environmental Monitoring*, 13, 3156

Pye, H.O.T., Liao, H., Wu, S., Mickley, L.J., Jacob, D.J., Henze, D.K., & Seinfeld, J.H. (2009). Effect of changes in climate and emissions on future sulfate-nitrate-ammonium aerosol levels in the United States. *Journal of Geophysical Research: Atmospheres, 114*

Sutton, M.A., Tang, Y.S., Miners, B., & Fowler, D. (2001). A New Diffusion Denuder System for Long-Term, Regional Monitoring of Atmospheric Ammonia and Ammonium. *Water Air & Soil Pollution Focus, 1*, 145-156

Tørseth, K., Aas, W., Breivik, K., Fjæraa, A.M., Fiebig, M., Hjellbrekke, A.G., Lund Myhre, C., Solberg, S., & Yttri, K.E. (2012). Introduction to the European Monitoring and Evaluation Programme (EMEP) and observed atmospheric composition change during 1972–2009. *Atmos. Chem. Phys.*, *12*, 5447-5481

Warner, J.X., Dickerson, R.R., Wei, Z., Strow, L.L., Wang, Y., & Liang, Q. (2017). Increased atmospheric ammonia over the world's major agricultural areas detected from space. *Geophysical Research Letters*

Whitburn, S., Van Damme, M., Clarisse, L., Bauduin, S., Heald, C.L., Hadji-Lazaro, J., Hurtmans, D., Zondlo, M.A., Clerbaux, C., & Coheur, P.F. (2016). A flexible and robust neural network IASI-NH3 retrieval algorithm. *Journal of Geophysical Research: Atmospheres*, *121*, 6581-6599

Xu, W., Luo, X.S., Pan, Y.P., Zhang, L., Tang, A.H., Shen, J.L., Zhang, Y., Li, K.H., Wu, Q.H., Yang, D.W., Zhang, Y.Y., Xue, J., Li, W.Q., Li, Q.Q., Tang, L., Lv, S.H., Liang, T., Tong, Y.A., Liu, P., Zhang, Q., Xiong, Z.Q., Shi, X.J., Wu, L.H., Shi, W.Q., Tian, K., Zhong, X.H., Shi, K., Tang, Q.Y., Zhang, L.J., Huang, J.L., He, C.E., Kuang, F.H., Zhu, B., Liu, H., Jin, X., Xin, Y.J., SHi, X.K., Du, E.Z., Dore, A.J., Tang, S.,

Collett Jr, J.L., Goulding, K., Sun, Y.X., Ren, J., Zhang, F.S., & Liu, X.J. (2015). Quantifying atmospheric nitrogen deposition through a nationwide monitoring network across China. *Atmospheric Chemistry and Physics*, *15*, 12345-12360

---

## Author Response (AR2)

Dear Dr. Frank Dentener,

Please find below our itemized responses to the referees' comments. We have addressed all the comments, and incorporated the comments/suggestions in the revised manuscript.

Thank you very much for your consideration.

Sincerely,

Xiuying Zhang and Lei Liu

On behalf of all co-authors

**Response to Referee #1**

The authors appreciate the valuable suggestions given by Referee #1 for improving the quality of the manuscript. Detailed responses to each comment are given below (in blue).

Some smaller issues:

Figure 4/5 explain what is meant with: "fertilizer plus N manure"? Emissions/N-application rates?

We mean it from N application. We have added it in the captions of Fig. 4 and 5.

Figure 8 label with regions are quite clogged. Improve presentation way.

We have revised Fig. 8 as follows:

[Figure]

**Fig. 8** MODIS fire counts in 2014. (a) Spatial distributions of MODIS fire counts. (b) Monthly variations of fire counts and surface NH$_3$ concentrations in biomass burning regions in China, the US and Europe.